# Topologically enhanced exciton transport

Joshua J. P. Thompson [1] ✉, Wojciech J. Jankowski [2], Robert-Jan Slager [2,3] & Bartomeu Monserrat [1]

Excitons dominate the optoelectronic response of many materials. Depending on the time scale and host material, excitons can exhibit free diffusion, phonon-limited diffusion, or polaronic diffusion, and exciton transport often limits the efficiency of optoelectronic devices such as solar cells or photodetectors. We demonstrate that topological excitons exhibit enhanced diffusion in all transport regimes. Using quantum geometry, we find that topological excitons are generically larger and more dispersive than their trivial counterparts, promoting their diffusion. We apply this general theory to organic polyacene semiconductors and show that exciton transport increases up to fourfold when topological excitons are present. We also propose that non-uniform electric fields can be used to directly probe the quantum metric of excitons, providing a rare experimental window into a basic geometric feature of quantum states. Our results provide a new strategy to enhance exciton transport in semiconductors and reveal that mathematical ideas of topology and quantum geometry can be important ingredients in the design of next-generation optoelectronic technologies.

Excitons, Coulomb-bound electron-hole pairs, dominate the optoelectronic response of a multitude of semiconductors[1–3]. Prominent examples include organic[4,5] and low-dimensional[6–8] semiconductors, each a vast and versatile family of compounds which host excitons with large binding energies that can reach hundreds of milielectronvolts[4,9]. The formation, dynamics, lifetime, and transport of excitons dictate the efficiency of a host of technological applications, from solar cells[10,11] and light-emitting diodes[12,13], to biosensors[14,15]. From a material perspective, the chemical and structural diversity available in the design of organic and low-dimensional semiconductors allows fine-tuning of the electronic and excitonic properties for customised device applications[16].

Despite their promise, one of the key limitations of organic semiconductors is the low mobility of excitons[9,17,18]. For example, low exciton mobility has been shown to inhibit the efficiency of organic semiconductor based solar cells[19] since excitons decay before being extracted. As another example, in some organic systems the fission of optically active singlet excitons into pairs of optically inactive triplet excitons could help boost efficiency beyond the Shockley-Queisser limit[11], but the diffusion of these triplets is even slower than that of singlets[18], and again exciton transport is a limiting factor. Other schemes, such as organic co-crystals[20,21] and organic-inorganic interfaces[22–24], again suffer from exciton mobility limitations.

In this work, we propose topology as a new avenue to enhance exciton transport. The topology of electrons is well-established[25–31], leading to remarkable transport properties such as quantum Hall phenomena[32]. A natural question to ask is whether topological ideas can be extended to excitons, which are starting to be explored in two-dimensional van der Waals layered materials[33,34], organic semiconductors[35], and idealised models[36,37]. In this context, a recent remarkable result concerns the topologically-induced non-trivial Riemannian geometry of exciton wavefunctions. Specifically, it has been demonstrated that exciton quantum geometry, phrased in terms of the quantum metric[38,39], provides a lower bound on the centre-of-mass spread $\xi$ of excitons[35]:

$$\xi^2 \geq \frac{a^2 P_{\text{exc}}^2}{4},$$ (1)

where $a$ is the lattice parameter of the crystal and $P_{\text{exc}}$ is an excitonic topological invariant protected by crystalline inversion symmetry[28–31]. The relationship between topology, quantum geometry, and exciton

[1]Department of Materials Science and Metallurgy, University of Cambridge, Cambridge, UK. [2]Theory of Condensed Matter Group, Cavendish Laboratory, University of Cambridge, Cambridge, UK. [3]Department of Physics and Astronomy, The University of Manchester, Manchester, UK. ✉e-mail: jjt56@cam.ac.uk

properties is general and can be applied to different topological invariants in different dimensions and in different material platforms[40]. For example, in a one-dimensional setting, exemplified by organic polyacenes[35,41,42], the excitonic topology can be characterised through a topological invariant $P_{exc} \in \mathbb{Z}_2$, associated with the first Stiefel-Whitney characteristic class $w_1 \in \mathbb{Z}_2$ that reflects the unorientability of an excitonic band[35,43,44].

Qualitatively, the bound in Eq. (1) implies that topological excitons are more delocalised, and can be larger, than their trivial counterparts. In this work, we exploit this key insight to demonstrate that topological excitons exhibit enhanced transport compared to their trivial counterparts. We demonstrate enhanced exciton transport in all regimes, ranging from free exciton diffusion at femtosecond timescales to phonon-limited and polaronic diffusion at longer picosecond timescales. We also illustrate these general results in a family of organic polyacene crystals, where we find that topological excitons exhibit a four-fold increase in their transport compared to their trivial counterparts. Overall, our work establishes topology as a new avenue for improving optoelectronic technologies.

## Results

### Topological excitons: model and materials

To explore the role of topology on exciton transport, we focus on a one-dimensional system that has recently been predicted to host topological excitons[35]. In this setting, single-particle electron properties are described by the Su-Schrieffer-Heeger (SSH) model[45,46]:

$$H = -t_1 \sum_j c_{B,j}^\dagger c_{A,j} - t_2 \sum_j c_{B+1,j}^\dagger c_{A,j} + \text{h.c.}, \qquad (2)$$

with $c_{B,j}^\dagger / c_{A,j}$ being the creation/annihilation operators for the electrons at sublattices $A$, $B$, in unit cell $j$, and alternating hopping parameters $t_1$ and $t_2$. The topological phase realising topological edge states corresponds to $t_2 > t_1$, and the trivial phase corresponds to $t_2 < t_1$[45,46].

From these single-particle electron and hole states, we then describe the exciton properties using the Wannier equation[24,35,47], which directly incorporates the electron-hole Coulomb interaction. The solution of the Wannier equation yields exciton bands $E_{\nu Q}$ associated with exciton states $|\psi_{\nu Q}^{exc}\rangle$, where $\nu$ is the band index and $Q$ is the exciton centre-of-mass momentum. We also introduce $|u_{\nu Q}^{exc}\rangle$ as the cell-periodic part of the excitonic Bloch state $|\psi_{\nu Q}^{exc}\rangle = e^{iQR}|u_{\nu Q}^{exc}\rangle$, where $R = (r_e + r_h)/2$ is the centre-of-mass position of the exciton, with electron position $r_e$ and hole position $r_h$. The topology of excitons in one-dimensional centrosymmetric semiconductors can be captured by a $\mathbb{Z}_2$ invariant $P_{exc}$, which can be directly obtained from the excitonic states[35].

A material realisation of this model is provided by polyacene chains composed of $n$-ring acene molecules, where $n = 3, 5, 7$, linked by a carbon-carbon bond on the central carbon atoms. Illustrative examples, polyanthracene ($n = 3$) and polypentacene ($n = 5$), are shown in Fig. 1. These polyacenes exhibit a topologically trivial exciton phase with $P_{exc} = 0$ for $n = 3$, and a topological phase with $P_{exc} = 1$ for $n = 5, 7$[35]. Following our previous work[35], we found that the excitons inherit their non-trivial topology from the underlying electronic topologies rather than through interaction effects[36]. The quasi-1D nature of these crystals and the weak dielectric screening of organic molecules lead to large excitonic binding energies, which, combined with the large band gaps, ensures that these topological excitons dominate the optical response.

We emphasise that the model and materials described above are for illustrative purposes only, and the key findings of this work are generally applicable to the transport of topological excitons in any material and dimension. When extending to higher dimensions and more complex structures, finding accurate tight-binding models will

become more challenging. However, established first-principles computational tools for electronic Wannierisation such as those regularly employed to describe electronic topology[39], will still be sufficient here. We also point out that recent advances in excitonic Wannierisation[48] could assist in the calculations presented here, from fully first-principles methods.

Throughout this work, we do not consider the role of defects or interfaces, which would lead to additional scattering mechanisms[49] or, if pronounced enough, localised states[50–52]. We assume a sufficiently large and clean system such that the optoelectronic behaviour is determined by the bulk excitons. Interestingly, some recent studies have shown defect-induced enhancements to electronic geometry[53], and how these could translate to the exciton picture will be the topic of future work.

### Free exciton propagation

Upon photoexcitation, excitons diffuse freely at femtosecond timescales[54–56]. The exciton diffusion constant is given by (see SM):

$$D_\nu = \frac{1}{2\hbar} \left\langle \frac{\partial^2 E_{\nu Q}}{\partial Q^2} \right\rangle + \frac{1}{\hbar} \sum_{\mu \neq \nu} \langle \Delta_Q^{\mu\nu} g_{xx}^{\mu\nu}(Q) \rangle, \qquad (3)$$

where $E_{\nu Q}$ is the exciton energy dispersion for band $\nu$, $\Delta_Q^{\mu\nu} = E_\mu(Q) - E_\nu(Q)$ is the energy difference between the pair of exciton bands $\mu$ and $\nu$, and $g_{xx}^{\mu\nu}(Q) = \langle \partial_Q u_{\mu Q}^{exc} | u_{\nu Q}^{exc} \rangle \langle u_{\nu Q}^{exc} | \partial_Q u_{\mu Q}^{exc} \rangle$ is the excitonic multiband quantum metric. The exciton diffusivity in Eq. (3) has two contributions: the first term arises from the energy dispersion, and the second term arises from the quantum geometric properties of the associated exciton states. In other SSH systems, such as polyacetylene, it has been shown that photoexcited excitons decay quickly into dark excitonic states with long lifetimes[57]. In experiments, the exciton diffusion can be observed over time scales longer than the typical radiative decay timescales of bright excitons, due to both the presence of long-lived dark states and the presence of a decreasing but still observable population of bright states[54].

We next show that the geometric term in the exciton diffusivity of Eq. (3) leads to enhanced transport for topological excitons. Starting with the flat band limit, the contribution from the exciton energy dispersion vanishes, as $\partial^2 E_{\nu Q}/\partial Q^2 = 0$. Therefore, the exciton diffusion comes entirely from the geometric contribution. The geometric contribution scales according to $\Delta_Q^{\mu\nu} g_{xx}^{\mu\nu}(Q) \propto 1/\Delta_Q^{\mu\nu}$ (see SM), and in the flat band limit, focusing on the lowest exciton band, we can approximate the geometric contribution to the diffusivity as $D_{\nu=1} \approx \frac{\Delta}{\hbar} \sum_{\mu \neq 1} \langle g_{xx}^{\mu\nu}(Q) \rangle$, where $\Delta$ is the smallest $Q$-independent gap from the band. We can then define the Brillouin zone average quantum metric $\langle g_{xx}^\nu(Q) \rangle$ associated with exciton band $\nu$ by tracing over the interband contributions according to $\langle g_{xx}^\nu(Q) \rangle = \sum_{\mu \neq \nu} \langle g_{xx}^{\mu\nu}(Q) \rangle$, and we obtain that the diffusivity in the flat band limit amounts to:

$$D_{\nu=1} \approx \frac{\Delta}{\hbar} \langle g_{xx}^{\nu=1}(Q) \rangle. \qquad (4)$$

Using the bound $\xi^2 \geq \frac{a^2 P_{exc}^2}{4}$ from Eq. (1), and noting that the exciton centre-of-mass spread is related to the metric according to $\xi^2 = \langle g_{xx}^{\nu=1}(Q) \rangle$[35], we identify a lower bound on the geometric contribution to the exciton diffusivity:

$$D_{\nu=1} \geq \frac{\Delta}{\hbar} \frac{a^2 P_{exc}^2}{4}. \qquad (5)$$

Therefore, diffusive exciton transport in the lowest exciton band is directly impacted by the underlying exciton topology: the geometric contribution to the exciton diffusivity exhibits a lower bound for topological excitons ($P_{exc} = 1$) but no bound for trivial excitons ($P_{exc} = 0$). This is a direct consequence of the lower bound on the

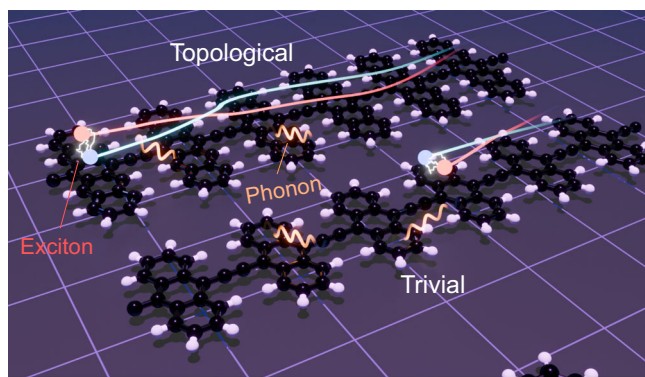

**Fig. 1 | Topologically enhanced exciton transport.** Topologically-enhanced diffusive transport of excitons (blue/red) with inversion symmetry-protected topological $\mathbb{Z}_2$ invariant $P_{exc}$ in the presence of phonons (wiggly orange lines). Due to exciton-phonon interactions, the propagating excitons in topological excitonic band can be scattered, dephased, and the diffusive transport of the excitons can be further altered with non-uniform electric fields introducing a controllable forcing. We show that non-trivial excitonic quantum geometry can be manifested in all these transport features.

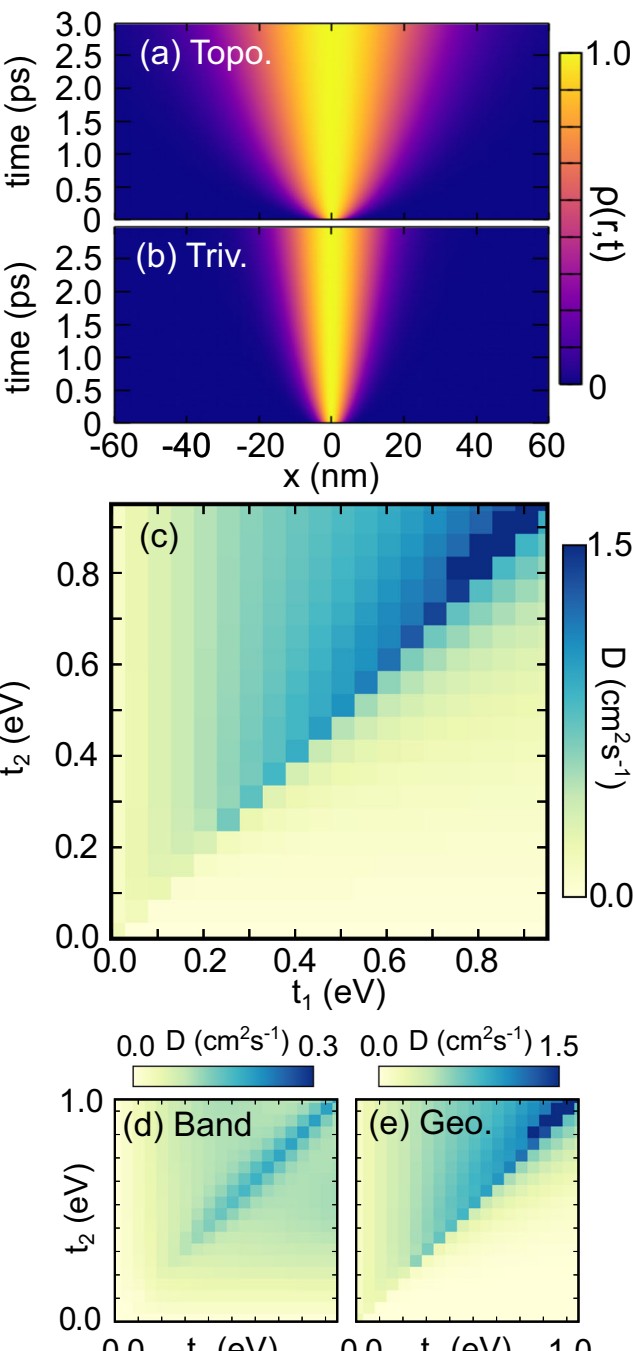

exciton centre-of-mass spread, as determined by quantum geometry, which makes topological excitons larger and therefore facilitates diffusion. In some organic materials, exciton transport is driven by long-range dipole-mediated hopping[58] due to the absence of band-driven transport and strong molecular dipoles. Crucially, our findings state that the lower bound sets a minimum diffusivity on the band-driven exciton transport in these materials, provided the excitons are topological, meaning that the exciton transport will also be enhanced even in systems where the hopping transport would otherwise dominate. In the polyacene chains, the Wannier-like nature of the excitons means these longer range exciton transfers are negligible.

Moving to the general dispersive case, both topological and trivial excitons will have equivalent contributions from the band dispersion to the diffusivity. Therefore, we can generally claim that topological excitons in the diffusive regime exhibit enhanced transport compared to their trivial counterparts.

To numerically illustrate the above results, we consider polypentacene as an example of a material hosting topological excitons with $t_2 > t_1$. We construct an initial exciton wavepacket, formed around the photoexcitation spot, and we calculate the subsequent exciton diffusion that leads to the spatial spread depicted in Fig. 2a. To test the importance of the topology-bound geometric contribution, we also consider the scenario in which the values of the hopping parameters are swapped, so that $t_2 < t_1$ and we are in the trivial regime. The corresponding exciton diffusion is depicted in Fig. 2b. The two scenarios have the same band dispersion, leading to the same first term in Eq. (3). However, the topological exciton diffuses more rapidly, a consequence of the geometric term in the diffusivity, which is bounded from below for topological excitons. These results explicitly demonstrate that exciton diffusion is enhanced in polypentacene driven by the underlying exciton topology.

More generally, Fig. 2c presents the diffusion constant as a function of $t_2$ and $t_1$ allowing us to demonstrate the wide applicability of our results. For any pair $\{t_1, t_2\}$, if $t_2 > t_1$ (topological excitons), then the diffusion constant is significantly larger than for the equivalent pair with $t_1 > t_2$ (trivial). When $t_1$ and $t_2$ are significantly different, the resulting diffusivities can differ by several orders of magnitude. The band contribution to the diffusion for topological and trivial excitons is equivalent (Fig. 2d), peaking at $t_1 - t_2$, where the electron and exciton band structures become most dispersive. In contrast, the geometric contribution is distinctly larger for the topological excitons compared to trivial ones (Fig. 2e). Non-zero topological transport in the flat band

**Fig. 2 | Enhanced free exciton diffusion.** Free diffusion of (**a**) topologically non-trivial (Topo.) and (**b**) trivial (Triv.) excitons. The diffusivity of topologically non-trivial excitons is bounded from below by the excitonic $\mathbb{Z}_2$ invariant. Parameters for $n = 5$ polypentacene are used, with the DFT predicted combination of intracell ($t_1$) and intercell ($t_2$) hoppings employed in (**a**) while the order is flipped in (**b**), to directly ascertain the impact of topology. **c** Exciton diffusion constant as a function of $t_1$ and $t_2$ with $t_2 > t_1$ ($t_2 < t_1$) representing topological and trivial excitons respectively[35]. Breakdown of the contribution to the exciton diffusion, shown in (**c**), by (**d**) exciton band dispersion and (**e**) exciton geometry (Geo.).

limit can be seen by comparing the $t_1 = 0$ or $t_2 = 0$ limit in Fig. 2d–e for the topological and trivial excitons, respectively. Additional material-specific data is shown in the Supplemental Material (SM) Fig. S1, which further elucidates the importance of the geometric contribution, particularly for polyheptacene, which has a significantly flatter band structure.

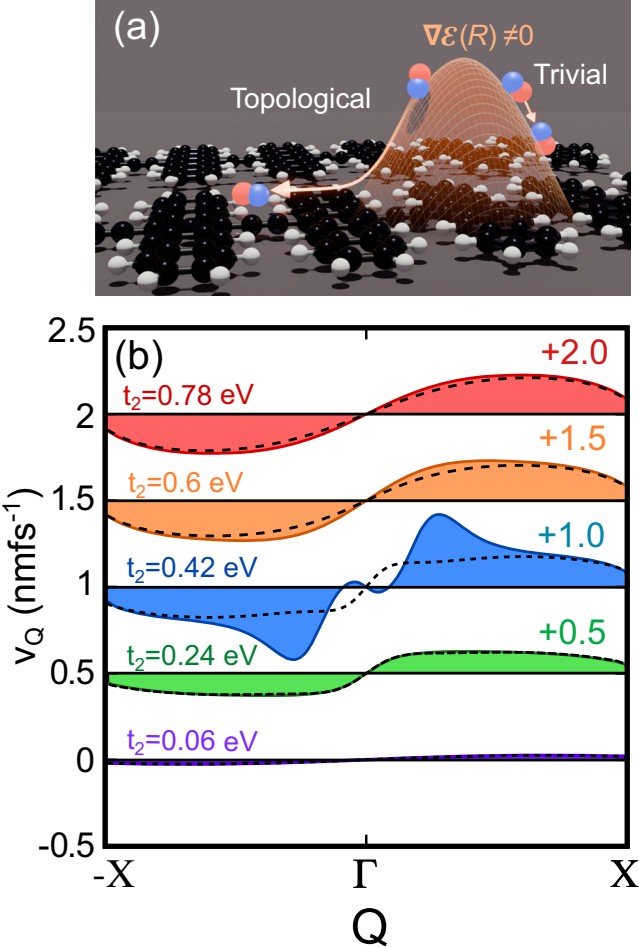

**Fig. 3 | Non-uniform electric fields. a** Schematic of non-uniform electric field, $\nabla\mathcal{E}(R)\neq0$, on a polypentacene crystal. The quantum metric of the topological exciton leads to a larger force due to the electric field (orange) compared to the trivial case. **b** Electric field induced tuning of the exciton group velocity $\bar{v}_Q$ of the lowest exciton band for different values of $t_2$ and fixed $t_1 = 0.33$ eV. Our extracted value of $t_2$ for polypentacene from DFT is 0.52 eV. The solid coloured lines (purple to red) show the exciton group velocity with an applied electric field while the dashed lines show the corresponding velocity in the absence of an applied electric field. The velocity plots with increasing $t_1$ are offset by 0.5 eV for clarity.

## Exciton transport in non-uniform electric fields

We next explore driven exciton transport under non-uniform electric fields, which we demonstrate can be used to directly probe the exciton quantum geometry. The exciton group velocity $\langle v_{\nu Q}\rangle$ associated with band $\nu$ is given in one dimension by (see Methods):

$$\langle v_{\nu Q}\rangle = \langle v_{\nu Q}^0\rangle - \sum_{\mu\neq\nu}\frac{e^2}{\hbar^2}\partial_Q\left(\frac{g_{xx}^{\mu\nu}(Q)}{\Delta_Q^{\mu\nu}}\right)(\langle r\rangle\cdot\nabla_R\mathcal{E}(R))^2, \quad (6)$$

where $\langle v_{\nu Q}^0\rangle$ is the free exciton group velocity and $\nabla_R\mathcal{E}(R)$ is the applied electric field gradient which couples to the electron-hole distance $\langle r\rangle$. According to Eq. (6), the total exciton group velocity has a contribution from the free exciton group velocity $\langle v_{\nu Q}^0\rangle$, and a contribution from the quantum metric derivatives. In one dimension, the latter can be described by the Christoffel symbols $\Gamma_{xxx}^{\mu\nu}(Q)=\frac{1}{2}\partial_Q g_{xx}^{\mu\nu}(Q)$. Overall, an exciton moving in a non-uniform electric field experiences a force, leading to either acceleration or deceleration of the exciton, and a modulation of the exciton group velocity.

The geometric contribution to the exciton group velocity in Eq. (6) depends on the energy difference $\Delta_Q^{\mu\nu}$ between bands $\mu$ and $\nu$. This

dependence can be suppressed by increasing dielectric screening, for example through strongly polar substrates, such that $\Delta_Q^{\mu\nu}\approx\Delta$ can be made approximately uniform over the exciton Brillouin zone. In this regime, the non-linear exciton transport in non-uniform electric fields is directly given by the quantum geometric Christoffel symbols.

In Fig. 3a, we schematically show the impact of an applied non-uniform electric field on exciton transport, where topological excitons experience an enhanced transport. Quantitatively, we calculate the exciton group velocity for polypentacene numerically using Eq. (6), and additional calculations where we vary $t_2$ freely are shown. For simplicity we set the electric field gradient to be constant $\nabla_R\mathcal{E}(R) = 0.1$ V/nm². Figure 3b shows the excitonic group velocity $\langle v_{\nu Q}\rangle$ modulated by a non-uniform electric field (coloured, shaded) at different values of $t_2$ for a fixed value $t_1= 0.3$ eV. The group velocity $v_Q$ in the absence of an external field is shown with the dashed lines. In the trivial regime, $\langle v_{\nu Q}\rangle\approx\langle v_{\nu Q}^0\rangle$ due to the vanishing quantum metric, $g_{xx}^{\mu\nu}\approx0$, and vanishing variations thereof, $\Gamma_{xxx}^{\mu\nu}\approx0$. The topological regime $(t_2 > t_1)$ shows a more complex behaviour. At small finite $Q$, the exciton diffusion is slowed down by the electric field with $\langle v_{\nu Q}\rangle\ll\langle v_{\nu Q}^0\rangle$ and even shows an opposite sign. At larger $Q$, the force induced by the non-uniform electric field on the topological excitons becomes larger, leading to a huge enhancement of the excitonic group velocity. This effect is most significant for $t_2$ reasonably close to $t_1$ within the range $t_2 < 0.5$ eV. For larger $t_2$, the quantum metric contribution shrinks, owing to the smaller excitons[35] such that the group velocity with and without electric field begins to converge again, see the red curve Fig. 3b.

Qualitatively, the distinct response of topological and trivial excitons under a non-uniform electric field can again be related to their different centre-of-mass localisations and relative sizes. Trivial excitons have a smaller size, and therefore are less subject to electric field gradients. The quantum metric in the momenta conjugate to the relative electron-hole position $r$, and the centre-of-mass coordinates $R$, precisely reflect the corresponding spreads and localisations of excitons[35] (see Methods).

## Phonon-limited exciton diffusion

Following free exciton diffusion at femtosecond timescales, excitons experience phonon-limited diffusion at picosecond timescales[59,60]. In this regime, the exciton diffusion is given by:

$$D_{\rm ph} = \sum_{Q,\nu}\frac{\langle v_{\nu Q}^2\rangle}{\Gamma_{\nu Q}}\frac{e^{-E_{\nu Q}/k_{\rm B}T}}{\mathcal{Z}}, \quad (7)$$

where $v_{\nu Q}$ is the exciton group velocity, $\Gamma_{\nu Q}$ is the exciton-phonon scattering rate, $E_{\nu Q}$ is the exciton band energy, $k_{\rm B}$ is the Boltzmann constant, $T$ is temperature, and $\mathcal{Z}$ is the partition function. The role that topology and quantum geometry play on phonon-limited exciton diffusion depends on the interplay between the exciton group velocity and exciton-phonon scattering rates featuring in Eq. (7).

Starting with the exciton group velocity, topological excitons exhibit enhanced velocity magnitudes (see Methods):

$$\langle v_{\mu Q}^2\rangle = \langle v_{\mu Q}\rangle^2 + \frac{1}{\hbar^2}\sum_{\nu\neq\mu}|\Delta_Q^{\mu\nu}|^2 g_{xx}^{\mu\nu}(Q). \quad (8)$$

Here, the second term represents the geometric contribution that enhances the squared magnitude of the group velocity of topological excitons.

In terms of exciton-phonon scattering rates, the key microscopic quantities are the exciton-phonon scattering matrix elements $\mathcal{D}_{Qq\beta}^{\mu\nu}$ which describe the scattering from an initial exciton $(\nu, Q)$ into a final exciton $(\mu, Q + q)$ mediated by a phonon $(\beta, q)$ of momentum $q$ and energy $\hbar\omega_{\beta q}$. In turn, the exciton-phonon matrix elements can be written in terms of individual electron-phonon scattering matrix

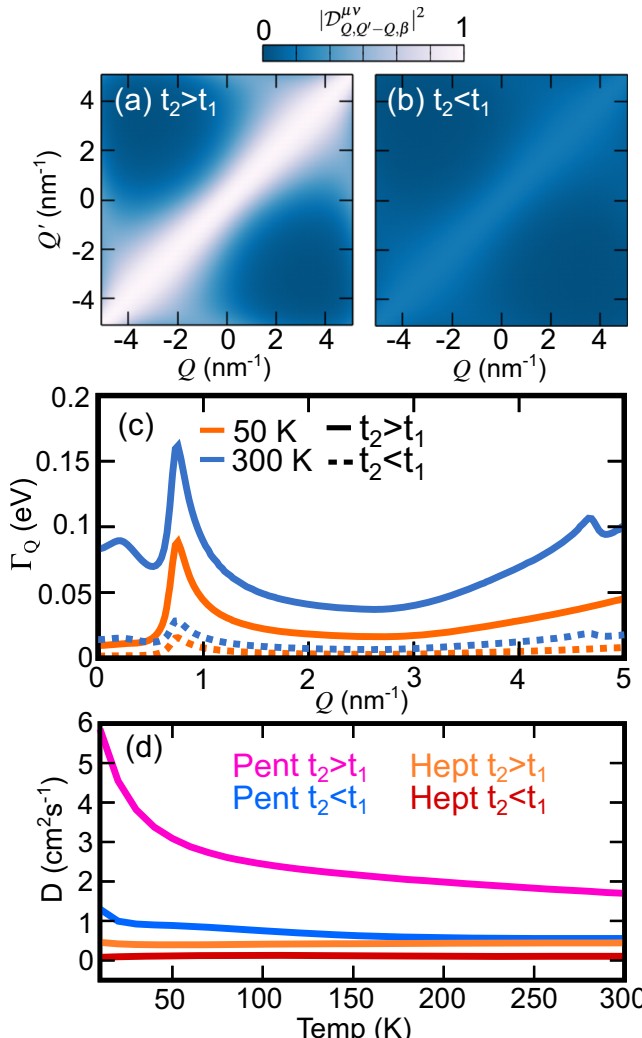

**Fig. 4 | Topology-enhanced phonon-mediated transport in polyacene chains.**
**a**–**b** Exciton-phonon matrix elements resolved in initial $Q$ and final $Q'$ excitonic
centre-of-mass momentum in Polypentacene. **c** Phonon-induced exciton dephasing as function of initial momentum $Q$ at 50 K (orange) and 300 K (blue) in polypentacene. The trivial and topological exciton dephasings are shown by the solid and dashed lines, respectively. **d** Phonon-mediated exciton diffusion for trivial (blue) and topological (pink) excitons in polypentacene (Pent) as a function of temperature (Temp). For comparison, we show the phonon-mediated exciton diffusion for polyheptacene (Hept) for trivial (red) and topological (orange) excitons.

elements $g_{k q \beta}^{m n}$ modulated by the exciton envelope function (see Methods). The electron-phonon scattering matrix elements describe the scattering from an initial electron (hole) $(n, k)$ into a final electron (hole) $(m, k + q)$, mediated by a phonon $(\beta, q)$. Topological electrons were previously found to significantly contribute to the electron-phonon coupling underpinned by $g_{k q \beta}^{m n}$ through electronic quantum geometric terms[61]. As a consequence, topological electrons enhance exciton-phonon coupling matrix elements $\mathcal{D}_{Q q \beta}^{\mu \nu}$, and we confirm this numerically as shown in Fig. 4a–b.

The preceding discussion implies that topological electrons will enhance the resulting exciton-phonon scattering matrix elements, but not all topological electrons lead to topological excitons. Topological excitons can arise from obstructed electrons and holes[35], and in this scenario the topology-enhanced electron-phonon scattering matrix elements will result in topologically enhanced exciton-phonon matrix elements. These, in turn, will lead to enhanced exciton-phonon scattering rates $\Gamma_{\nu Q}$. However, unobstructed electrons and holes can also give rise to topological excitons due to the electron-hole envelope

contribution[35,36]. In this second scenario, there is no enhancement of the electron-phonon scattering matrix elements, resulting in topological excitons that exhibit no enhancement in the exciton-phonon scattering rates $\Gamma_{\nu Q}$.

Overall, we end up with two scenarios. In the first scenario, the diffusion of topological excitons is enhanced when the underlying electrons and holes are trivial, driven by the topologically-driven enhancement of the exciton group velocity $v_{\nu Q}$. In the second scenario, corresponding to topological excitons with underlying topological electrons and holes, both the exciton group velocity $v_{\nu Q}$ and the exciton-phonon scattering rates $\Gamma_{\nu Q}$ are enhanced. The diffusivity of Eq. (7) depends on the ratio $v_{\nu Q}/\Gamma_{\nu Q}$, and therefore the diffusion of topological excitons in this scenario may be enhanced or suppressed. In the numerical example below, the enhancement of the group velocity dominates and the topological excitons exhibit enhanced transport.

To illustrate these results numerically, we consider the topological excitons in polyacenes. Polyacenes exhibit topological excitons with underlying topological electrons and holes. This is the only regime we can explore as there are no known material candidates hosting topological excitons with underlying trivial electrons and holes. In Fig. 4a–b, we show the exciton-phonon scattering matrix elements from an initial state $Q$ to a final state $Q'$ for polypentacene $(t_2 > t_1)$ and compare it to the trivial counterpart where the values of the hopping parameters are swapped $(t_2 < t_1)$. We use dimensionless units, as we are interested in the impact of the topology rather than the absolute values of the matrix elements. We observe different couplings for different momenta, depending on the topology associated with the Zak phases of the electronic and hole states comprising the excitons, with the peak intensities being dictated by the quantum geometry of individual electrons and holes, as well as their momentum-dependent interaction. We find that the $Q/Q'$ dependence on the exciton-phonon coupling is the same in both the trivial and topological case, however, the magnitude is significantly enhanced in the topological case, as expected from the discussion above.

One way to probe the impact of phonon scattering is via the exciton dephasing $\Gamma_Q = \sum_\nu \Gamma_{\nu Q}$. When $Q = 0$, the dephasing corresponds to the non-radiative lifetime of the lowest exciton state. In Fig. 4c, we present the calculated exciton dephasing as a function of momentum at 50 K (orange) and 300 K (blue) for polypentacene (solid lines) and its trivial counterpart (dashed lines). The dephasing depends on the population of phonons, which increases as a function of temperature. As such the dephasing at 300 K is significantly larger than that at 50 K. Irrespective of temperature, the dephasing is larger in the topological case, which can be understood by the larger magnitude of the exciton-phonon matrix elements of the topological regime compared to the trivial one. The excitonic dispersions themselves are almost identical, so any density of states effects in the allowed scattering channels[59] are approximately equivalent for both trivial and topological exciton dephasing. As a result, the same qualitative features are observed in the dephasing curves for both topological and trivial exictons at high and low temperatures. An initial increase in the dephasing can be observed at small $Q$, characterised by the emission of acoustic phonons scattering back to the $Q = 0$ state or at larger temperatures, absorption of phonons. This gives rise to a distinct bump feature[59] between $Q = 0.1$ nm$^{-1}$ and $Q = 0.5$ nm$^{-1}$. At around $Q = 0.8$ nm$^{-1}$, the exciton energy difference compared to $Q = 0$ nm$^{-1}$ corresponds to the optical phonon energy. As a result intraband optical phonon relaxation becomes possible leading to a sharp increase in the dephasing. At larger momentum $Q$ such relaxation remains possible, however, the exciton density of states at higher-momentum states is lower, leading to an overall decrease in the dephasing. At very large $Q > 3$ nm$^{-1}$, the exciton band flattens (cosine-like) leading to an increase in the excitonic density of states and a corresponding increase in scattering channels. As a result, peak is seen in the 300 K dephasing at $Q = 4.7$

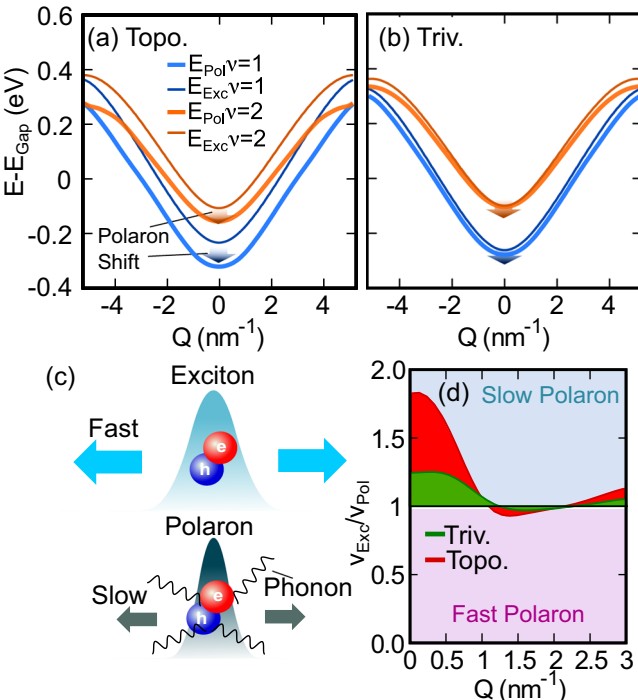

**Fig. 5 | Transport of topological exciton-polarons.** Exciton-polaron and exciton band dispersion, relative to band gap ($E_{Gap}$) for (**a**) topological and (**b**) trivial excitons. The lowest (next lowest) polaron energy is shown in blue (orange). A clear polaron shift is observed in both cases. **c** Schematic of reduced transport of exciton-polarons compared to bare excitons. **d** Ratio of free exciton to exciton-polaron group velocities in polypentacene at 300K for trivial (Triv.) and topological (Topo.) regimes, shown in green and red, respectively. The blue region indicates mass enhancement and slower exciton-polarons while the pink region indicates mass reduction and faster excitons.

nm$^{-1}$, but an equivalent peak is not present in the 50 K results as the thermal occupation of optical phonons is very small in the latter case.

We further calculate the phonon-limited exciton diffusion coefficients using Eq. (7) and report the results in Fig. 4d. In this example, the competition between the geometric contribution to the excitonic group velocity and to the enhanced exciton-phonon coupling leads to an overall enhancement of the diffusion in the topological regime. Taking solely the band contribution to the exciton velocity, the increased exciton-phonon dephasing associated with topological excitons leads to topological excitons diffusing about four times more slowly than trivial excitons at all temperatures, see Fig. S1. However, taking the exciton band geometry into account, leads to an increase in the exciton group velocity at low $Q$ in both the trivial and topological regime. While a fairly modest increase in the case of trivial excitons, the vastly enhanced exciton metric in the topological regime leads to a large increase in the exciton group velocity of the low $Q$, and yet highly populated, $Q = 0$ states. As a result the exciton diffusion is much larger in the topological regime, even despite the enhanced exciton-phonon coupling which increases the scattering term. The temperature dependence in Fig. 4d reflects this interpretation, with low temperatures corresponding to an increase in the relative population of low momentum excitons which have a large group velocity enhancement. The reduced exciton-phonon coupling at low temperatures adds to this behaviour and we observe a monotonic decrease in the exciton diffusion in both trivial and topological excitons. To demonstrate the generality of the topological enhancement to the phonon-mediated exciton transport, we also compare the trivial and topological regime for polyheptacene. Due to the flatter bands, the exciton transport is reduced in both regimes. For the real topological system, the raw diffusion values at room temperature (300 K) are 1.76 cm$^2$/s and 0.44

cm$^2$/s for polypentacene and polyheptacene, respectively. In contrast, the trivial states have diffusion constants of 0.61 cm$^2$/s and 0.103 cm$^2$/s for bond-reordered polypentacene and polyheptacene, respectively. Here, the topological diffusion constant at room temperature is around 3 times larger than the trivial case for polypentacene, compared to around 4.5 for polyheptacene. This difference stems from the larger percentage contribution of the geometric to the excitonic group velocity due to the flatness of the exciton band. This suggests that geometry of the excitons, rather than the geometry of the electron-phonon coupling, is the dominant practical factor for the transport enhancements. Given the characteristic energy scales driving these phenomena, we expect analogous effect interplays in the other optoelectronic systems realising dynamics dominated by excitons.

**Polaronic effects**

When the interaction between excitons and phonons becomes sufficiently large, excitons can become localised by the lattice[62,63], becoming heavier and undergoing slower transport. These exciton-polarons have been studied extensively[64–66], and are particularly relevant in organic systems where their formation hinders the already limited energy transfer across organic optoelectronic devices[67]. Hence, understanding the transport of excitons in this strong coupling regime is crucial.

We calculate the exciton band dispersion for polypentacene as renormalised by exciton-polarons at 300 K by treating the exciton-phonon interaction self-consistently (see Methods). The renormalised exciton-polaron dispersion for polypentacene is shown in Fig. 5a, with the trivial counterpart shown in Fig. 5b. In both cases, polaron formation results in an energy shift and in a decrease in the group velocity, but notably the topological exciton-polaron exhibits a larger energy shift and a larger reduction in velocity with a correspondingly increased mass, which we attribute to the topology-driven increase in the exciton-phonon interactions. Figure 5d shows the ratio of the free excitonic to the polaronic group velocities. We find the usual low-momentum decrease of the exciton-polaron velocity ($v_{Exc}/v_{Exc-Pol} > 1$) in both the trivial (green) and topological (red) cases, corresponding to a polaron velocity around 70% of that of the free exciton.

Experimentally, the formation of exciton polarons will lead to a redshift of the excitonic resonance energy[68] in the spectrum of absorbed/emitted light on a picosecond timescale. Importantly, the band topology of the exciton-polarons is the same as that of the bare excitons, and we note that at large $Q$ the topological exciton-polaron bands are close in energy but do not cross, analogous to the bare exciton case. Our results show that the band contribution to the polaron velocity is reduced in the topological case, however, the same metric contribution to the exciton transport holds, given that the polaron bands do not cross and possess the same underlying metric.

## Discussion

Overall, our results show that the topologically-bounded localisation properties of excitons dramatically affect their transport properties. Compared to their trivial counterparts, topological excitons sustain faster transport. The enhanced transport of topological excitons is expected to be experimentally trackable in the polyacenes[41], where the underlying electronic topology has already been observed. Experimentally, the dynamics of exciton transport can be visualised with time-resolved photoluminescence[54,69] or with transient absorption[70].

We show that topological excitons also experience stronger electron-phonon coupling-driven exciton-phonon coupling, compared to their trivial counterparts. This observation respects the expected enhancement of electron-phonon coupling of the constituent electrons and holes that host non-trivial quantum geometry[61]. As discussed earlier, the stronger exciton-phonon coupling experienced by topological excitons results in higher dephasing rates, but we find that these are not sufficient for the topological excitons to violate

the original quantum geometric bounds of the free exciton propagation, as compared to the trivial excitons. Similarly, the metric-enhanced transport of topological excitons still persists in the polaronic regime. These findings, accounting for the presence of physical effects present in all semiconducting materials, show that our diagnosis of quantum geometric manifestations on excitons should persist under experimental conditions.

Finally, we stress that the exciton transport properties and the associated exciton quantum geometry and topology can be controlled using an appropriate dielectric environment[35], chemical modifications, and temperature, which modifies the population of the exciton (and phonon) states. Therefore, our findings provide a general quantum mechanical formalism and mathematical insights to theoretically understand the experimentally controllable geometric manifestations due to excitonic topologies, as reflected in the discussed excitonic transport in semiconductor materials.

We have demonstrated that the transport of topological excitons is significantly enhanced compared to that of trivial excitons. This discovery arises from the lower bound that the centre-of-mass excitonic quantum geometry sets on the exciton localisation, making topological excitons larger and therefore more mobile. We have shown that enhanced topological exciton transport holds in sub-picosecond free transport regime and in the picosecond phonon-limited and polaronic transport regimes. Additionally, we have illustrated these discoveries in a family of polyacene organic semiconductors. Our results are general, and we expect that exciton topology can be exploited to enhance the transport properties of a wide variety of semiconductors for applications in optoelectronic devices.

## Methods
### Exciton quantum geometry
We consider an exciton state associated with exciton band $\nu$ and centre-of-mass momentum $\mathbf{Q}$:

$$|\psi_{\nu\mathbf{Q}}^{\text{exc}}\rangle = \sum_{\mathbf{k}} \psi_{\nu\mathbf{Q}}(\mathbf{k}) e^{i\mathbf{k}\cdot\mathbf{r}} |u_{\mathbf{k}+\mathbf{Q}/2}^{\text{e}}\rangle |u_{-\mathbf{k}+\mathbf{Q}/2}^{\text{h}}\rangle, \qquad (9)$$

where $\psi_{\nu\mathbf{Q}}(\mathbf{k})$ is the envelope function capturing the electron-hole correlation, $\mathbf{r} = \mathbf{r}_{\text{e}} - \mathbf{r}_{\text{h}}$ is the relative electron-hole distance with the associated relative momentum $\mathbf{k}$, and $|u_{\mathbf{k}+\mathbf{Q}/2}^{\text{e}}\rangle$ and $|u_{-\mathbf{k}+\mathbf{Q}/2}^{\text{h}}\rangle$ are the single-particle electron and hole states. Exploiting translational symmetry, we can also write the exciton state as:

$$|\psi_{\nu\mathbf{Q}}^{\text{exc}}\rangle = e^{i\mathbf{Q}\cdot\mathbf{R}} |u_{\nu\mathbf{Q}}^{\text{exc}}\rangle, \qquad (10)$$

where $\mathbf{R} = \frac{\mathbf{r}_{\text{e}}+\mathbf{r}_{\text{h}}}{2}$ is the centre-of-mass coordinate, and the exciton state satisfies Bloch's theorem with the cell-periodic part given by:

$$|u_{\nu\mathbf{Q}}^{\text{exc}}\rangle = e^{-i\mathbf{Q}\cdot\mathbf{R}} \sum_{\mathbf{k}} e^{i\mathbf{k}\cdot\mathbf{r}} \psi_{\nu\mathbf{Q}}(\mathbf{k}) |u_{\mathbf{k}+\mathbf{Q}/2}^{\text{e}}\rangle |u_{-\mathbf{k}+\mathbf{Q}/2}^{\text{h}}\rangle. \qquad (11)$$

The quantum Riemannian geometry associated with exciton states was originally introduced in ref. 35. The quantum geometric tensor in the centre-of-mass coordinates $(\mathbf{R})_i \sim i\partial_{Q_i}$ is given by:

$$\mathcal{Q}_{ij}^{\text{exc},\nu}(\mathbf{Q}) = \langle \partial_{Q_i} u_{\nu\mathbf{Q}}^{\text{exc}} | (1-\hat{P}_{\nu\mathbf{Q}}) | \partial_{Q_j} u_{\nu\mathbf{Q}}^{\text{exc}} \rangle, \qquad (12)$$

where $\hat{P}_{\nu\mathbf{Q}} = |u_{\nu\mathbf{Q}}^{\text{exc}}\rangle\langle u_{\nu\mathbf{Q}}^{\text{exc}}|$ is a projector onto the exciton band of interest. Its real part, the quantum metric, is given by:

$$g_{ij}^{\text{exc},\nu}(\mathbf{Q}) = \frac{\langle \partial_{Q_i} u_{\nu\mathbf{Q}}^{\text{exc}} | (1-\hat{P}_{\nu\mathbf{Q}}) | \partial_{Q_j} u_{\nu\mathbf{Q}}^{\text{exc}} \rangle + \langle \partial_{Q_j} u_{\nu\mathbf{Q}}^{\text{exc}} | (1-\hat{P}_{\nu\mathbf{Q}}) | \partial_{Q_i} u_{\nu\mathbf{Q}}^{\text{exc}} \rangle}{2}, \qquad (13)$$

and it relates to the centre-of-mass spread of excitons $\langle(\mathbf{R}-\langle\mathbf{R}\rangle)^2\rangle$. Importantly, the relation between the exciton spread and the quantum metric can be exploited to reconstruct the exciton quantum metric in transport experiments that involve freely propagating and driven excitons, as we show in the main text.

### Free exciton diffusion
In the free diffusive regime, following Fick's second law, the temporal and spatial evolution of the exciton density can be expressed as:

$$\rho_{\nu}(x,t) = \frac{N_0}{\sqrt{2\pi(2D_{\nu}t+\sigma_{\text{Ini}}^2)}} \exp\left[\frac{-(x-x_{\text{Ini}})^2}{2(2D_{\nu}t+\sigma_{\text{Ini}}^2)}\right], \qquad (14)$$

where $N_0$ is the initial number of generated excitons in excitonic band $\nu$, and for well-localised excitons we have $\sigma_{\text{Ini}}^2 \approx \xi^2$, where $x_{\text{Ini}}$ and $\sigma_{\text{Ini}}$ are the initial excitation centre and broadening, respectively.

In the following, we show that the exciton diffusivity $D_{\nu}$ in band $\nu$ is fully captured by the centre-of-mass quantum metric of the excitons $g_{xx}(Q)$. By mapping the quantum dynamics of free excitons in the diffusive regime to Fokker-Planck Gaussian propagation in one spatial dimension, we obtain that $\sigma^2(t) = \sigma_{\text{Ini}}^2 + 2D_{\nu}t$, with $D_{\nu} = \frac{\hbar}{2m_{\nu}^*} \rightarrow \langle g_{xx}^{\nu}(Q)\rangle$. The full derivation is detailed in the Supplemental Material (SM), but briefly, we map the density time-evolution equation to the Focker-Planck equation, in order to connect the diffusivity to the effective excitonic mass $m_{\nu}^*$. Furthermore, we utilise the Hellmann-Feynman theorem to derive the relation between the effective excitonic mass ($m_{\nu}^*$) and the quantum-geometry in the centre-of-mass momentum space. As a result, we find that the diffusivity of excitons in band $\nu$ is given by:

$$D_{\nu} = \frac{1}{2\hbar} \left\langle \frac{\partial^2 E_{\nu Q}}{\partial Q^2} \right\rangle + \frac{1}{\hbar} \sum_{\mu\neq\nu} \left\langle \Delta_Q^{\mu\nu} g_{xx}^{\mu\nu}(Q) \right\rangle, \qquad (15)$$

where $E_{\nu Q}$ is a dispersion of band $\nu$, and the averages are taken with respect to the Brillouin zone spanned in the $Q$ momentum space parameter (see also SM).

### Driven exciton transport under non-uniform electric fields
We consider exciton transport driven by an external non-uniform electric field gradient, complementary to the field gradients realisable internally within the system[71]. Semiclassically, interacting electrons and holes satisfy the equation of motion[72]:

$$\dot{\mathbf{k}}_{\text{e/h}} = -\nabla_{\mathbf{r}_{\text{e/h}}} U(\mathbf{r}_{\text{e}}-\mathbf{r}_{\text{h}}) \mp e\mathcal{E}(\mathbf{r}_{\text{e/h}}), \qquad (16)$$

where $U(\mathbf{r}_{\text{e}}-\mathbf{r}_{\text{h}})$ is the electron-hole interaction potential. This implies that the centre-of-mass exciton momentum $\mathbf{Q} = \mathbf{k}_{\text{e}} + \mathbf{k}_{\text{h}}$ satisfies an equation of motion with a position-dependent external force $\mathbf{F}(\mathbf{R})$:

$$\begin{aligned} \dot{\mathbf{Q}} &= e[\mathcal{E}(\mathbf{r}_h) - \mathcal{E}(\mathbf{r}_e)] = e[\mathcal{E}(\mathbf{R}+\mathbf{r}/2) - \mathcal{E}(\mathbf{R}-\mathbf{r}/2)] \\ &= e\langle\mathbf{r}\rangle \cdot \nabla_{\mathbf{R}}\mathcal{E}(\mathbf{R}) = \mathbf{F}(\mathbf{R}). \end{aligned} \qquad (17)$$

In this expression, we use $\mathbf{R} = (\mathbf{r}_{\text{e}} + \mathbf{r}_{\text{h}})/2$, $\mathbf{r} = \mathbf{r}_{\text{e}} - \mathbf{r}_{\text{h}}$ and that to first order, $\mathcal{E}(\mathbf{R} \pm \mathbf{r}/2) = \mathcal{E}(\mathbf{R}) \pm \mathbf{r}/2 \cdot \nabla_{\mathbf{R}}\mathcal{E}(\mathbf{R}) + O(\mathbf{r}^2)$.

Physically, the quantum geometric coupling to $\dot{\mathbf{Q}}$ can be related to the renormalised exciton energies. Consider a perturbation coupling to the centre of mass of the exciton $\Delta H = -\mathbf{R} \cdot \mathbf{F}(\mathbf{R})$, where $\mathbf{R}$ is a position operator projected onto an excitonic band. For the off-diagonal elements, we have $\langle \psi_{\mu\mathbf{Q}}^{\text{exc}} | \mathbf{R} | \psi_{\nu\mathbf{Q}}^{\text{exc}} \rangle = i\langle u_{\mu\mathbf{Q}}^{\text{exc}} | \nabla_{\mathbf{Q}} u_{\nu\mathbf{Q}}^{\text{exc}} \rangle$. At second order in perturbation theory, and assuming that the exciton bands are

non-degenerate, we obtain the following energy corrections:

$$
\tilde{E}_{\nu\mathbf{Q}} = E_{\nu\mathbf{Q}} - \sum_{\mu\neq\nu} \frac{|\langle\psi_{\nu\mathbf{Q}}^{\mathrm{exc}}|\Delta H|\psi_{\mu\mathbf{Q}}^{\mathrm{exc}}\rangle|^2}{E_{\mu\mathbf{Q}} - E_{\nu\mathbf{Q}}} = E_{\nu\mathbf{Q}}
$$
$$
- \sum_{\mu\neq\nu} \frac{\mathbf{F}^{\mathrm{T}}(\mathbf{R})\cdot\langle\psi_{\nu\mathbf{Q}}^{\mathrm{exc}}|\mathbf{R}|\psi_{\mu\mathbf{Q}}^{\mathrm{exc}}\rangle\langle\psi_{\mu\mathbf{Q}}^{\mathrm{exc}}|\mathbf{R}|\psi_{\nu\mathbf{Q}}^{\mathrm{exc}}\rangle\cdot\mathbf{F}(\mathbf{R})}{E_{\mu\mathbf{Q}} - E_{\nu\mathbf{Q}}},
$$

(18)

which in terms of the excitonic quantum metric, we can rewrite as

$$
\tilde{E}_{\nu Q} = E_{\nu Q} - \sum_{\mu\neq\nu} \frac{g_{xx}^{\mu\nu}(Q)}{E_{\mu Q} - E_{\nu Q}} F(R)F(R),
$$

(19)

for a one-dimensional system. Here, $\tilde{E}_{\nu Q}$ is the excitonic energy renormalised by the coupling to external force fields $F(R)$. Denoting $\Delta_Q^{\mu\nu} = E_{\mu Q} - E_{\nu Q}$, and substituting $F(R) = \hbar\dot{Q} = e\langle r\rangle\cdot\nabla_R\mathcal{E}(R)$, we arrive at:

$$
\langle v_{\nu Q}\rangle = \frac{1}{\hbar}\partial_Q\tilde{E}_{\nu Q} = \langle v_{\nu Q}^0\rangle - \sum_{\mu\neq\nu}\frac{e^2}{\hbar^2}\partial_Q\left(\frac{g_{xx}^{\mu\nu}(Q)}{\Delta_Q^{\mu\nu}}\right)\left(\langle r\rangle\cdot\nabla_R\mathcal{E}(R)\right)^2,
$$

(20)

where $\langle v_{\nu Q}^0\rangle = \frac{1}{\hbar}\partial_Q E_{\nu Q}$ is the free exciton velocity.

The above result implies that varying the electric field gradient in transport experiments allows the reconstruction of the derivatives of the exciton quantum metric. As mentioned in the main text, in the flat-band limit $\Delta_Q^{\mu\nu} \approx \Delta$, the Christoffel symbols $\Gamma_{xxx}^{\mu\nu} = \frac{1}{2}\partial_Q g_{xx}^{\mu\nu}(Q)$[73] can be directly accessed with this strategy. It should be noted that the size of the exciton, given by the average of the relative electron-hole coordinate $\langle r\rangle$, must be known to assess the magnitude of the net force $F(R)$ due to the electric field gradient $\nabla_R\mathcal{E}(R)$. Correspondingly, we compute the average size of the exciton that is relevant for the semiclassical equation of motion directly from the envelope function: $\langle r\rangle = \int_0^\infty \mathrm{d}r\, r|\psi_{\nu Q}(r)|^2$, where $\psi_{\nu Q}(r)$ is a Fourier transform of $\psi_{\nu Q}(k)$[35].

From the perspective of quantum geometry, we note that the derivatives of the quantum metric defining the Christoffel symbols can be in principle arbitrarily high due to the envelope contributions to the excitonic quantum metric[35], resulting in a nearly step-like character for $g_{xx}^{\mathrm{exc}}(Q)$ in the presence of singular non-Abelian Berry connections. Such singular behaviours of non-Abelian excitonic Berry connection are only to be expected for topological excitons. In the trivial phases with vanishing topological invariants, the Berry connection can be chosen to be globally smooth.

## Exciton group velocity

The group velocity term featuring in the phonon-limited exciton diffusion and in the exciton-polaron diffusion has a quantum geometric contribution. To derive it, we use a resolution of the identity in terms of excitonic states, $1 = \sum_\nu |u_{\nu Q}\rangle\langle u_{\nu Q}|$, and find that:

$$
\begin{aligned}
\langle v_{\mu Q}^2\rangle &= \frac{1}{\hbar^2}\langle u_{\mu Q}|(\partial_Q H_Q)^2|u_{\mu Q}\rangle \\
&= \frac{1}{\hbar^2}\sum_\nu\langle u_{\mu Q}|\partial_Q H_Q|u_{\nu Q}\rangle\langle u_{\nu Q}|\partial_Q H_Q|u_{\mu Q}\rangle \\
&= \frac{1}{\hbar^2}\left(\partial_Q E_{\mu Q}\right)^2 + \frac{1}{\hbar^2}\sum_{\nu\neq\mu}\langle u_{\mu Q}|\partial_Q H(Q)|u_{\nu Q}\rangle \\
&\quad\times \langle u_{\nu Q}|\partial_Q H(Q)|u_{\mu Q}\rangle \\
&= \langle v_{\mu Q}\rangle^2 + \frac{1}{\hbar^2}\sum_{\nu\neq\mu}|\Delta_Q^{\mu\nu}|^2 g_{xx}^{\mu\nu}(Q),
\end{aligned}
$$

(21)

with $\Delta_Q^{\mu\nu} = E_{\mu Q} - E_{\nu Q}$, which allows the multiband exciton quantum metric elements $g_{xx}^{\mu\nu}(Q)$ to modify the phonon-mediated diffusion via interband velocity matrix elements. Intuitively, the latter determine the variance of the velocity operator. On substituting the exciton quantum metric-dependent $\langle v_{\mu Q}^2\rangle$ for $D_{\mathrm{ph}}$, we observe that the geometric contribution enhances the phonon-mediated diffusion of the topological excitons.

## Exciton-phonon coupling

In this section, we consider the connection between exciton-phonon coupling (ExPC) matrix elements[74] and quantum geometry. The electron-phonon coupling (EPC) Hamiltonian can be written as:

$$
H_{\mathrm{el-ph}} = \sum_{k,m,n,q,\beta} g_{kq\beta}^{mn}\hat{a}_{mk+q}^\dagger\hat{a}_{nk}\left(\hat{b}_{\beta,q} + \hat{b}_{\beta,-q}^\dagger\right),
$$

(22)

where $\hat{a}_{nk}^{(\dagger)}$ is the annihilation (creation) operator for an electron in band $n$ and momentum $k$. Similarly, $\hat{b}_{\beta q}^{(\dagger)}$ is the annihilation (creation) operator for a phonon with mode $\beta$ and momentum $q$. The coupling between electrons and phonons is quantified by the general interband matrix elements $g_{kq\beta}^{mn}$. The EPC matrix elements $g_{kq\beta}^{mn}$ for electron-phonon scattering between bands $m$ and $n$, in terms of electron Bloch states read:

$$
g_{kq\beta}^{mn} = \sqrt{\frac{\hbar}{2M\omega_{q\beta}}}\langle u_{mk}|\partial_q H|u_{nk+q}\rangle,
$$

(23)

where $H = \sum_i E_i|\psi_i\rangle\langle\psi_i|$ is the many-body Hamiltonian of the system combining the electron and phonon degrees of freedom, $|\psi_i\rangle$ are the many-body ground and excited eigenstates with energies $E_i$, and $M$ is the ionic effective mass. In the case of the polyacenes, the effective mass is dominated by the heavier carbon atoms.

To make our discussion concrete, we will consider a two-band model a conduction band $c$ and a valence band $v$. This regime is applicable to the polyacene chains discussed in the main text. Correspondingly, we define $g_{kq\beta c} \equiv g_{kq\beta}^{cc}$ and $g_{kq\beta v} \equiv g_{kq\beta}^{vv}$. We further define a pair operator basis as:

$$
\hat{a}_{ck+q}^\dagger\hat{a}_{ck} = \sum_l \hat{P}_{k+q,l}^\dagger\hat{P}_{l,k}, \quad \hat{a}_{vk+q}^\dagger\hat{a}_{vk} = \sum_l \hat{P}_{l,k+q}\hat{P}_{k,l}^\dagger,
$$

(24)

and we rewrite the electron-phonon coupling in this basis as:

$$
H_{\mathrm{el-ph}} = \sum_{k,l,q,\beta}\left(g_{kq\beta c}\hat{P}_{k+q,l}^\dagger\hat{P}_{l,k} + g_{kq\beta v}\hat{P}_{l,k+q}\hat{P}_{k,l}^\dagger\right)\left(\hat{b}_{\beta,q} + \hat{b}_{\beta,-q}^\dagger\right).
$$

(25)

We can then rewrite the Hamiltonian in the exciton basis:

$$
H_{\mathrm{ex-ph}} = \sum_{Q,q,\beta}\mathcal{D}_{Qq\beta}^{\mu\nu}\hat{X}_{Q+q}^{\mu\dagger}\hat{X}_Q^\nu\left(\hat{b}_{\beta,q} + \hat{b}_{\beta,-q}^\dagger\right),
$$

(26)

$$
\begin{aligned}
\mathcal{D}_{Qq\beta}^{\mu\nu} = \sum_k \Big( & g_{kq\beta c}\psi_{Q+q\mu}\left(k - \tfrac{1}{2}Q + \tfrac{1}{2}q\right)\psi_{Q\nu}^*\left(k - \tfrac{1}{2}Q\right) \\
& - g_{kq\beta v}\psi_{Q+q\mu}\left(k + \tfrac{1}{2}Q - \tfrac{1}{2}q\right)\psi_{Q\nu}^*\left(k + \tfrac{1}{2}Q\right)\Big),
\end{aligned}
$$

(27)

where the electron-phonon coupling (EPC) matrix elements $g_{kq\beta v}$ reflect the quantum geometry of the underlying electrons and holes[61]. Contributions to ExPC explicitly originate from the free-particle EPC matrix elements ($g_{kq\beta v}$) and from the overlaps of excitonic envelope functions $\psi_Q(k)$ governed by the excitonic quantum geometry that was defined in the previous section. In the excitons considered in our work, $\psi_Q(k)$ is almost identical for both inverse ratios $t_1/t_2$ and $t_2/t_1$, yet the EPC part, $g_{kq\beta v}$, changes significantly. To understand this relation, we note that by considering the Hamiltonian derivatives $\partial_q H$ within a Gaussian approximation for effective hopping parameters $t_{ij}(x)$ under a phonon displacement of magnitude $x$, $t_{ij}(x) = t_{ij}e^{-\gamma x^2}$, following Ref. 61, the geometric contributions to EPC matrix elements can be approximated as:

$$
|g_{kq\beta v}^{\mathrm{geo}}|^2 \approx \frac{\hbar}{2M\omega_{q\beta}}\left(\gamma\left(\sum_\mu g_{ij}^{\mu\nu,\mathrm{elec}}(\mathbf{k}) + \dots\right)\right).
$$

(28)

In the above, consistently with Ref. [61], we recognise the presence and the significance of an electronic multiband quantum metric $g_{ij}^{\mu\nu,\text{elec}}(k) = \text{Re}\left\langle \partial_{k_i} u_{\nu k} | 1 - \hat{P}_\mu | \partial_{k_j} u_{\nu k}\right\rangle$, with $\hat{P}_\mu = |u_{\mu k}\rangle\langle u_{\mu k}|$ a projector onto the electronic band with index $\mu$. On combining with the ExPC equation, this demonstrates the importance of quantum metric contributions to the exciton-phonon coupling, in particular contributed by the electronic quantum metric. Importantly, the electrons with the non-trivial topological invariant, will significantly contribute with the highlighted geometric terms to the enhancement of both EPC and ExPC in the topological (obstructed) electronic phase.

In the calculations for exciton dephasing, diffusion, and polaron shift, we define realistic values of $\gamma$ for acoustic and optical phonons according to previous calculations/experiments on oligoacene semiconductors[47], obtaining realistic values for the exciton linewidths. We note, however, that our focus is primarily on the relative difference between different transport phenomena in topological and trivial regimes rather than predicting the absolute values.

### Exciton-polaron formation

The full Hamiltonian describing a system hosting excitons and phonons can be written as

$$H = H_{\text{ex},0} + H_{\text{ph},0} + H_{\text{ex-ph}}. \tag{29}$$

To describe the impact of phonons on the excitonic properties, we define a new polaronic Hamiltonian which absorbs the impact of the exciton-phonon coupling into the single-particle energies. Following Ref. [65], we define a polaronic transformation:

$$S = \sum_{Q,q\nu,\beta} \mathcal{D}_{Qq\beta}^{\mu\nu} \left( \frac{1}{E_{\nu Q+q} - E_{\mu Q} + \hbar\omega_{q\beta}} \hat{b}_{\beta,-q}^\dagger \right.$$
$$\left. + \frac{1}{E_{\nu Q+q} - E_{\mu Q} - \hbar\omega_{q\beta}} \hat{b}_{\beta,q} \right) \hat{X}_{Q+q}^{\nu\dagger} \hat{X}_Q^\mu, \tag{30}$$

which allows us to rewrite the Hamiltonian as:

$$\tilde{H} = H_{\text{ex},0} + H_{\text{ph},0} - \frac{1}{2}\left[S, H_{\text{ex-ph}}\right]. \tag{31}$$

On solving the commutator, we arrive at the following Hamiltonian:

$$\tilde{H} = H_{\text{ex},0} + H_{\text{ph},0} - \sum_{Q,q\nu,\beta} |\mathcal{D}_{Qq\beta}^{\mu\nu}|^2$$
$$\left( \frac{n_q^\beta + 1}{E_{\nu Q+q} - E_{\mu Q} + \hbar\omega_{q\beta}} + \frac{n_q^\beta}{E_{\nu Q+q} - E_{\mu Q} - \hbar\omega_{q\beta}} \right) \hat{X}_Q^{\mu\dagger} \hat{X}_Q^\mu. \tag{32}$$

Here, $n_q^\beta$ describes the population of phonons in mode $\beta$ and momentum $q$, which we model using a Bose-Einstein distribution assuming a thermalised phonon bath[59,65]. The Hamiltonian $\tilde{H}$ can be solved for the phonon-interaction corrected excitonic envelopes $\tilde{\psi}_{\mu Q}(\mathbf{k})$ on achieving self-consistency with the calculated self-energies $\Sigma_{\mu Q}$, the associated dephasing rates $\Gamma_{\mu Q} = \text{Im}\,\Sigma_{\mu Q}$, and the given exciton-phonon interaction matrix elements $\mathcal{D}_{Qq\beta}^{\mu\nu}$. Namely, we have $\tilde{E}_{\mu Q} = E_{\mu Q} - \text{Re}\,\Sigma_{\mu Q}$, with:

$$\text{Re}\,\Sigma_{\mu Q} = -\lim_{\delta_0 \to 0} \Re \sum_{q,\nu,\beta} |\mathcal{D}_{Qq\beta}^{\mu\nu}|^2$$
$$\left( \frac{n_q^\beta + 1}{E_{\nu Q+q} - E_{\mu Q} + \hbar\omega_{q\beta} + i\Gamma_Q + i\delta_0} + \frac{n_q^\beta}{E_{\nu Q+q} - E_{\mu Q} - \hbar\omega_{q\beta} + i\Gamma_Q + i\delta_0} \right). \tag{33}$$

We observe a clear polaron shift, as shown in Fig. 5 of the main text, and a minor renormalisation of the excitonic effective mass. The excitonic mass renormalisation arises from the Feynman diagrams

associated with the coupling of the virtual phonon cloud to the excitons[75]. Finally, on differentiating the polaron-renormalised band energy $\tilde{E}_{\mu Q}$, we obtain:

$$\langle \tilde{v}_{\mu Q} \rangle = \frac{1}{\hbar} \partial_Q \tilde{E}_{\mu Q}, \tag{34}$$

the polaron-renormalised exciton group velocities $\langle \tilde{v}_{\mu Q} \rangle$. Here, implicitly, the derivatives of the matrix elements $\mathcal{D}_{Qq\beta}^{\mu\nu}$ entering the self-energy $\Sigma_{\mu Q}$ that satisfies a self-consistency condition, allow the excitonic quantum geometry to affect the renormalised exciton transport in the presence of a phonon cloud.

Having considered the effects of the virtual phonons on the exciton masses and velocities, we moreover consider an expectation value $\langle \tilde{v}_{\mu Q}^2 \rangle$. Analogously as in the main text, this quantity enters the phonon-mediated diffusivity that accounts for a polaron shift $\tilde{D}_{\text{ph}}$, which is mediated by the temperature-dependent scattering of exciton-polarons from the thermally-populated phonons:

$$\tilde{D}_{\text{ph}} = \sum_{Q,\nu} \frac{\langle \tilde{v}_{\nu Q}^2 \rangle}{\Gamma_Q} \frac{e^{-\beta\tilde{E}_{\nu Q}}}{\mathcal{Z}}, \tag{35}$$

with thermodynamic $\beta = \frac{1}{k_B T}$, and $\mathcal{Z}$ a partition function for exciton-polaron states. Using a derivation analogous to that in Eq. (21) for the group velocity of excitons, we find that for the polaronic states we can write:

$$\langle \tilde{v}_{\mu Q}^2 \rangle = \langle \tilde{v}_{\mu Q} \rangle^2 + \frac{1}{\hbar^2} \sum_{\nu \neq \mu} |\tilde{\Delta}_Q^{\mu\nu}|^2 \tilde{g}_{xx}^{\mu\nu}(Q), \tag{36}$$

with $\tilde{\Delta}_Q^{\mu\nu} = \tilde{E}_{\mu Q} - \tilde{E}_{\nu Q}$, which allows the renormalised multiband exciton quantum metric elements $\tilde{g}_{xx}^{\mu\nu}(Q)$ to modify the phonon-mediated diffusion via interband velocity matrix elements. Intuitively, the latter determine the variance of the renormalised velocity operator. On substituting the exciton quantum metric-dependent $\langle \tilde{v}_{\mu Q}^2 \rangle$ for $\tilde{D}_{\text{ph}}$, we observe that the geometric contribution enhances the phonon-mediated diffusion of the topological exciton-polarons.

Finally, we note that in the presence of exciton-polaron corrections[65,76], the topology of excitons remains unaltered. Furthermore, the transport in the presence of a non-uniform electric field qualitatively overlaps with the calculation which did not involve the renormalisation with phonons. We show the corresponding results in Fig. 5 of the main text.

## Data availability
All datasets for the plots of this study are available upon request to the authors. All data is reproducible using the equations and input parameters outlined in the manuscript.

## Code availability
All codes and associated data are reproducible with information in the manuscript. All first-principles calculation input files are available upon request to the authors.

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

## Acknowledgements

The authors thank Richard Friend and Akshay Rao for helpful discussions. J.J.P.T. and B.M. acknowledge support from an EPSRC Programme Grant [EP/W017091/1]. W.J.J. acknowledges funding from the Rod Smallwood Studentship at Trinity College, Cambridge. R.-J.S. acknowledges funding from a New Investigator Award, EPSRC grant EP/W00187X/1, an EPSRC ERC underwrite grant EP/X025829/1, a Royal Society exchange grant IES/R1/221060, and Trinity College, Cambridge. B.M. also acknowledges support from a UKRI Future Leaders Fellowship [MR/V023926/1] and from the Gianna Angelopoulos Programme for Science, Technology, and Innovation. Calculations were performed using the Sulis Tier-2 HPC platform hosted by the Scientific Computing Research Technology Platform at the University of Warwick. Sulis is funded by EPSRC Grant [EP/T022108/1] and the HPC Midlands+ consortium.

## Author contributions

B.M. and R.-J.S. initiated the project. J.J.P.T. performed all numerical first-principles and excitonic calculations with inputs from B.M. J.J.P.T. derived microscopic theory with input from W.J.J. and B.M.; W.J.J. performed theoretical analysis of quantum geometry with inputs from R.-J.S. All figures and schematics were created by J.J.P.T. with input from the rest of the authors. All authors discussed the results and substantially contributed to the writing of the manuscript. The final form of the manuscript, including Methods and Supplementary Information, benefitted from input from all authors.

## Competing interests

The authors declare no competing interests.
