## [Transparent Peer Review file · Nature Communications]

Topologically enhanced exciton transport

Corresponding Author: Dr Joshua Thompson

Version 0:

Reviewer comments:

Reviewer #1

(Remarks to the Author)

The manuscript discusses how topology (or more specifically, quantum geometry) can affect exciton transport, with a focus on organic semiconductors. The authors demonstrate that it's possible to design materials where excitons exhibit topologically non-trivial states. These topological excitons are predicted to be larger and more delocalized than their trivial counterparts, leading to significantly enhanced diffusion and transport. The theory is applied to organic polyacene semiconductors, where the authors predict a four-fold increase in exciton transport. Furthermore, they propose an experimental method to probe these effects using non-uniform electric fields.

However, the current form of the manuscript is not suitable for publication in Nature Communications for the following reasons:

The overall discussion is thorough; however, I have a big concern about the starting point:

1. The authors used the SSH model to understand the properties of exciton transport. It is known that the excitons in the SSH model are only stable at a subpicosecond time scale and will eventually decay into solitons; see, for example, PhysRevLett.124.017401. This raises a question: to what extent is the authors' modeling valid, especially in the picosecond region (Fig. 2, for example)?

Besides, the claim that the theory is general could be potentially overclaiming:

2. A key part of the manuscript addresses transport at picosecond timescales, where exciton-phonon interactions are assumed to be dominant. The authors identify a competition between:

* Velocity Enhancement: Topological excitons have a higher group velocity due to their quantum geometry. This should increase diffusion.

* Scattering Enhancement: The same underlying properties enhance exciton-phonon coupling, leading to higher scattering rates (dephasing). This should decrease diffusion.

The authors conclude that velocity enhancement wins, leading to a net increase in transport for topological excitons.

However, this conclusion is based entirely on a numerical simulation for polypentacene. The authors even state that if they ignore the geometric velocity enhancement, the increased scattering would make topological excitons diffuse four times more slowly than trivial ones. This reveals that the central claim of enhanced transport in this important regime may not be a universal rule but a material-dependent outcome. This is a significant limitation on the generality of the findings.

Other than that, some technical details should be made clearer:

3. How does one get $\chi = \langle g_{xx}^{\nu=1}(Q) \rangle$ (between eq 4 and 5)? This seems an important equation to relate the diffusion constant and the topology of the exciton state.

4. A related question: Despite the fact that Eq. 5 gives a lower bound of the diffusion constant for the topological state in the flatband limit, it's worthwhile to compare the energy dispersion contribution and the quantum geometric contribution to the diffusion constant. If the quantum geometric contribution is a higher-order effect compared to the energy dispersion part, the bound becomes less useful.

5. What numerical method is used? Across the manuscript, it seems only the broad term "numerical simulation" is mentioned. Without the specific numerical methods and some detailed parameter specification, the reliability of the numerical results is doubtful.

6. The discussion on exciton-polarons—where excitons become "dressed" by lattice distortions—is somewhat contradictory. The text claims that topological excitons are more "robust to polaronic effects." However, the data shows that topological exciton-polarons exhibit a larger energy shift and a greater reduction in velocity compared to their trivial counterparts, precisely because of the stronger exciton-phonon coupling. The plot in Figure 5(d) also shows a more significant velocity

reduction ($V_{\text{Exc}} / V_{\text{Exc-Pol}} > 1$) for the topological case at low momentum. The claim of "robustness" seems to refer to the fact that the system's topological invariants are not destroyed by polaron formation, not that the transport properties are less degraded. This phrasing is confusing and could be misinterpreted.

Besides, another closely related question is not discussed:

7. How does disorder affect the transportation in such a system? Will it be so dominating that the enhancement of the topological excitation becomes negligible?

Reviewer #2

(Remarks to the Author)

In this work, the authors explore a novel approach to enhancing exciton transport by leveraging topological excitons in organic materials. The study is clearly a continuation of their previous work published in the same journal [DOI: 10.1038/s41467-025-59257-5] (below I refer to it as paper 1). The analysis presented is both innovative and timely and is likely to be of considerable interest to the community. However, I believe the manuscript would benefit significantly from a stronger connection to application-relevant questions, experimentally accessible observables, and common organic systems. As it stands, the work feels too technical for the broad readership of Nat. Commun., and I would encourage the authors to provide more context and motivation to make the manuscript accessible to a wider audience.

1) What are the requirements for the formation of topological excitons? It would be helpful to include a brief explanation of the key structural and electronic characteristics needed to realize such excitonic states or systems. While part of this discussion is included in the authors' previous work (paper 1), revisiting these points here would make the present study more self-contained and accessible.

2) Is it truly necessary to use quasi-metallic polymers (such as acenes connected by carbon-carbon bonds) to realize this topologically enhanced transport? In other words, does the system need to be very conductive in order to maintain topological features? I am asking this because in Fig. 4 the diffusion coefficient both in topological and trivial exciton are very large (at least one to two orders of magnitude larger than in common van der Waals organic crystals). In my view, this raises the question about whether the novel physics highlighted in this work is only applicable to quasi-metallic systems, rather than to a broader class of organic materials as the authors seem to suggest. Could the authors clarify this and add some discussion?

3) Regarding the electronic structure, I believe the quality and accuracy of their tight-binding SSH Hamiltonian stems from its ability to accurately fit the valence and conduction bands of the systems considered. However, in extending this approach to more complex systems, such as 2D and 3D materials or systems with more intricate band structures, do the authors think a TB Hamiltonian would be accurate enough? How do the authors envisage modifying the Hamiltonian? A more detailed discussion of these future directions would be valuable. At present, the manuscript only briefly touches on this with a general statement on page 2: "the key findings of this work are generally applicable to the transport of topological excitons in any material and dimension." A deeper and more concrete discussion of the broader applicability would strengthen the manuscript.

4) The SSH Hamiltonian in Eq. 2 appears to include only nearest-neighbor interactions, am I correct? However, excitonic couplings are generally long-range, and it is well known that the sign and magnitude of non-nearest-neighbor interactions can significantly affect both the spectroscopic and transport properties of molecular aggregates (see, for example, the work of Spano et al.). Could the authors comment on the potential role of long-range interactions in their model? Specifically, would the topological features and associated transport behavior be affected by the inclusion of such terms?

5) From my perspective is not quite clear how the author introduced temperature in their calculations and related polaronic effects. What are they computing exactly with electronic structure?

6) The authors mentioned that "Experimentally, the dynamics of exciton transport can be visualised with time-resolved photoluminescence [43, 56] or with transient absorption [57]." Is there any experimental evidence (maybe hidden) that demonstrates topological enhancement in materials, which the authors can refer more specifically to?

Reviewer #3

(Remarks to the Author)

Author Rebuttal letter:

Reviewer 1 (Remarks to the Author):

The manuscript discusses how topology (or more specifically, quantum geometry) can affect exciton transport, with a focus on organic semiconductors. The authors demonstrate that it's possible to design materials where excitons exhibit topologically non-trivial states. These topological excitons are predicted

to be larger and more delocalised than their trivial counterparts, leading to significantly enhanced diffusion and transport. The theory is applied to organic polyacene semiconductors, where the authors predict a four-fold increase in exciton transport. Furthermore, they propose an experimental method to probe these effects using non-uniform electric fields.

We cordially thank the Referee for taking their time to read our manuscript and their questions that helped to improve our manuscript.

However, the current form of the manuscript is not suitable for publication in Nature Communications for the following reasons:

Below we address each of the Referee's questions/concerns point-by-point.

The overall discussion is thorough; however, I have a big concern about the starting point: 1. The authors used the SSH model to understand the properties of exciton transport. It is known that the excitons in the SSH model are only stable at a subpicosecond time scale and will eventually decay into solitons; see, for example, PhysRevLett.124.017401. This raises a question: to what extent is the authors' modeling valid, especially in the picosecond region (Fig. 2, for example)?

The Referee asks about the timescales involved in our work and an apparent contradiction between the approximately ~ 100 femtosecond (fs) decay into solitons as described in [Phys. Rev. Lett. 124, 017401 (2020)], versus the transport in our work measured over a scale of picoseconds. Here, we clarify that that system explored in [Phys. Rev. Lett. 124, 017401 (2020)] is (trans)-polyacetylene, the prototypical SSH chain system. Following the nomenclature of [Phys. Rev. Lett. 124, 017401 (2020)], we note that in such systems there are two possible phases, namely "A and B" phases. These are energetically the same, as the lattice sites in polyacenes are chemically the same, and the difference arises from the location of the carbon-carbon double bond to be between (A-phase) or within (B-phase) unit cells. Experimentally, it was proposed that for long polymer chains, interfaces between these A and B phases can form, giving rise to soliton-like states.

To resolve the Referee's concern, we note that despite the identification of the femtosecond decay, the referenced paper actually disproves the existence of exciton-soliton splitting, further claiming that the photophysics of the material on the picosecond timescales is governed by dark excitons. In the conclusion, the paper explicitly states:

"In conclusion, depending on the excitation photon energy, both singlet excitons (binding energy of about ~ 0.33 eV and polarons are instantaneously photogenerated in $t\text{-(CH)}_x$. Importantly, no soliton-antisoliton pairs are photogenerated directly upon photon absorption. With these findings, the saga of photoexcitations in $t\text{-(CH)}_x$ has been solved. The photophysics of $t\text{-(CH)}_x$ in the ps time domain is governed by a dark exciton that lies below the allowed exciton state but otherwise is in agreement with other homopolar

1/12

π -conjugated polymers."

In other words, the measured (fs) decay time is instead attributed to relaxation of the optically excited exciton into an energetically lower dark state. For these dark exciton states, the lack of a radiative decay channel would lead to even longer diffusion lengths. Similar phenomena are observed in tungsten-based TMDs, where long diffusion lengths of (momentum)-dark excitons can be observed [2D Materials 8 015030 (2020)]. Our conclusions on the topologically-enhanced exciton transport apply to both bright and dark states as long as the underlying electron-hole bands are topological.

We also point out that our findings, including the enhanced exciton diffusion constants, exciton-phonon coupling, and non-uniform electric fields are considered in the time-independent regime, i.e., in the steady-state limit, and are valid under the formation of thermalised exciton distributions. As such, even with alternative relaxation mechanisms, which reduce the lifetime of excitons, the exciton transport within this characteristic decay time will still be enhanced and measurable. Even after times several times longer than the characteristic decay time, a non-zero population of excitons will exist and will be measurable. For example, spatiotemporal photoluminescence measurements, typically used to measure the diffusion of excitons, are often performed over significantly longer timescales (ps) than the radiative decay time (fs) [2D Materials 8, 015030 (2020), Nature Communications 11, 2035 (2020)].

To address the Referee's concern, which we answered above – we make the following changes in the revised main text:

In other SSH systems, such as polyacetylene, it has been shown that photoexcited excitons decay quickly into dark excitonic states with long lifetimes [Phys. Rev. Lett. 124, 017401 (2020)]. In experiments, the exciton diffusion can be observed over time scales longer than the typical radiative decay timescales of bright excitons, due to both the presence of long-lived dark states and the presence of a decreasing but

still observable population of bright states [2D Materials 8, 015030 (2020)].

We have now added the listed reference [Phys. Rev. Lett. 124, 017401 (2020)], as well as the related reference: [2D Materials 8, 015030 (2020)], in the revised manuscript.

Besides, the claim that the theory is general could be potentially overclaiming:

2. A key part of the manuscript addresses transport at picosecond timescales, where exciton-phonon interactions are assumed to be dominant. The authors identify a competition between:

* Velocity Enhancement: Topological excitons have a higher group velocity due to their quantum geometry. This should increase diffusion.

* Scattering Enhancement: The same underlying properties enhance exciton-phonon coupling, leading to higher scattering rates (dephasing). This should decrease diffusion.

The authors conclude that velocity enhancement wins, leading to a net increase in transport for topological excitons. However, this conclusion is based entirely on a numerical simulation for polypentacene. The authors even state that if they ignore the geometric velocity enhancement, the increased scattering would make topological excitons diffuse four times more slowly than trivial ones. This reveals that the central claim of enhanced transport in this important regime may not be a universal rule but a material-dependent

2/12

outcome. This is a significant limitation on the generality of the findings.

We thank the Referee for their comment. We agree that we originally made a rather general statement based on our simulation for polypentacene, with transferrability to the other SSH systems in mind. Below we clarify this statement in two directions: first, we extend the numerical analysis to an additional system; and second, we provide a general argument to show that in most systems we would expect topology to enhance exciton transport.

To strengthen our claim beyond the individual material realisation of polypentacene, we now also show data for polyheptacene, which is a distinct SSH system. Polyheptacene has a much flatter excitonic and electronic dispersion and hence the underlying excitonic quantum geometry will play a more dominant role due to the vanishing band contribution. We find that polyheptacene also shows larger topological exciton transport than the trivial counterpart, however the absolute magnitudes are lower owing to the flatter exciton bands. This can be seen clearly in the revised Fig. 1, which we include below for reference.

Figure 1. Updated diffusion Figure showing enhanced diffusion for both topological and trivialised polypentacene (pink and blue, respectively) and topological and trivialised polyheptacene (orange and red, respectively).

For the realistic topological systems – the raw diffusion values at room temperature (300 K) are 1.76 cm²/s and 0.44 cm²/s for polypentacene and polyheptacene, respectively. In contrast, the artificial trivial states (made by swapping the order of t₁ and t₂ with the same underlying electronic band structure) have diffusion constants of 0.61 cm²/s and 0.103 cm²/s for bond-reordered polypentacene and polyheptacene, respectively. The generality of this finding can also be tested through uniaxial strain as outlined in our previous work [Nature Communications 16, 4661 (2025)], allowing us to control the topology and hence geometry of our excitons.

We next consider a more general argument regarding the interplay between topology-enhanced exciton velocity contrasting with topology-enhanced exciton-phonon scattering rates. This comes down to whether the electronic geometry $\langle g_{\text{xelec}}(Q) \rangle$, which dictates the electron-phonon coupling, can outcompete the geometry stemming from the excitonic eigenstates, $\langle (g_{\text{v=1}} \times x(Q)) \rangle$, first defined in [Nature Communications 16, 4661 (2025)]. Our theory shows a lower bound on the excitonic geometry $\langle g_{\text{v=1}} \times x(Q) \rangle \geq 4$, which is similar to an equivalent bound on the electronic geometry $\langle g_{\text{xelec}}(Q) \rangle \geq 4$, see e.g., [arXiv:2502.02660]. As such, the relevant quantities are then the energy scales linking: (1) the excitonic geometry $\langle g_{\text{v=1}} \times x(Q) \rangle$

3/12

to the velocity enhancement, and (2) the electronic geometry $\langle g_{\text{xelec}}(Q) \rangle$ to the enhanced electron-phonon

and exciton-phonon couplings [Eq. (28)]. The exciton velocity enhancement scales the exciton geometry $v=1(Q)$ according to $(\Delta\mu v)^2/\hbar^2$ [Eq. (21)], where $\Delta\mu v$ is the energy separation between energy levels.

$[(g_{xx} Q Q$

Therefore, we need to compare the energy separation between exciton levels and the exciton-phonon μv

coupling energy scale. For the former, the typical excitonic energy level separation ΔQ in organics is of the order of hundreds of meV, and similarly for other systems with prominent excitons such as 2D semiconductors. By comparison, the typical energy scale associated with exciton-phonon coupling is of the order of tens of meV. Comparing these energy scales, we expect the larger geometric contribution for excitons to dominate in systems with strongly bound excitons such as organic and 2D materials.

Going back to our quantitative calculations, we find that excitonic geometry $[(g_{v=1} xx(Q))]$ enhances the

exciton velocity by a factor of around ~ 50 at low Q in the studied polyacenes (the factor decreases for large Q), with low Q excitons dominating in the exciton population, and hence dominating the transport. In contrast, we find around a factor of ~ 7 increase to the exciton-phonon scattering rate for topological excitons at low Q . This result is of the same order of magnitude as that reported by previous works on electron-phonon coupling, which found that electronic geometry can enhance electron-phonon scattering by factors of $\sim 2-3$ in graphene and MgB_2 , respectively [Nature Physics 20, 1262 (2024)].

We believe that the recipe to engineer the reverse behaviour (topologically-reduced phonon-mediated diffusion) can be realised in systems with low binding energy such as in three-dimensional inorganic μv^2

systems [AIP Advances 3, 11 (2013)], as the product of $(g_{v=1} xx(Q))$ and $(\Delta Q)/\hbar$ is suppressed, with

μv

$\Delta Q \leq 10$ meV. While such systems with low binding energies (both trivial and topological) have a universally weaker exciton-phonon coupling due to more weakly bound excitons, the electronic geometric contribution $[(g_{elec} xx(Q))]$

to the electron-phonon coupling does not depend on this. Alternatively, as discussed in Ref. [48], systems with vanishing energetic contribution to the electron-phonon coupling for a given phonon mode would lead to larger enhancements due to the geometric contributions. This would be a topic of future work, looking at the chemical design of systems to engineer large geometric contributions.

Finally, we note that, although not present in the organic systems discussed here, it is possible that topological/trivial excitons can be generated from trivial/topological electrons, respectively, as discussed in our previous work Ref. [31] and also in Ref. [21]. Here, we believe it is possible to disentangle the two effects, with either topological enhancement to the group velocity of excitons or topological enhancement to the electron-phonon coupling, but not the reverse.

In summary, for most optoelectronic systems realising dynamics dominated by excitons, we believe that, as supported by our calculations, and based on a more fundamental energetic argument above, the geometry of the excitons, rather than the geometry of the electron-phonon coupling, is the dominant practical factor.

We update the text accordingly to include a relevant discussion:

To demonstrate the generality of the topological enhancement to the phonon-mediated exciton transport, we also compare the trivial and topological regime for polyheptacene. Due to the flatter bands, the exciton transport is reduced in both regimes. For the real topological system, the raw diffusion values at room temperature (300 K) are $1.76 \text{ cm}^2/\text{s}$ and $0.44 \text{ cm}^2/\text{s}$ for polypentacene and polyheptacene,

4/12

respectively. In contrast, the trivial states have diffusion constants of $0.61 \text{ cm}^2/\text{s}$ and $0.103 \text{ cm}^2/\text{s}$ for bond-reordered polypentacene and polyheptacene, respectively. Here, the topological diffusion constant at room temperature is around 3 times larger than the trivial case for polypentacene, compared to around 4.5 for polyheptacene. This difference stems from the larger percentage contribution of the geometric to the excitonic group velocity due to the flatness of the exciton band. This suggests that geometry of the excitons, rather than the geometry of the electron-phonon coupling, is the dominant factor for the transport enhancements. Given the characteristic energy scales driving these phenomena, we expect analogous effect interplays in the other optoelectronic systems realising dynamics dominated by excitons.

We also updated the corresponding figure and caption.

Other than that, some technical details should be made clearer:

3. How does one get $\xi = \langle g_{v=1} xx(Q) \rangle$ (between eq 4 and 5)? This seems an important equation to relate the diffusion constant and the topology of the exciton state.

The relationship $\xi^2 = \langle gv=1$
xx (Q)) is a central non-trivial result of the cited Ref. [31], which arises analogously to the electronic localisation, on a basis transformation between the Bloch basis realizing the metric, and the Wannier (or coherent state) basis [Phys. Rev. B 56, 12847, Phys. Rev. B 62, 1666]. We agree with the Referee on the importance of this equation to relate the diffusion constant and the topology (manifested by geometry) of the exciton state.

The relation between the localisation length and diffusion, as underpinned by our bound, intuitively arises through the relations of: (1) localisation length geometrically amounting to the excitonic quantum metric, as shown in Ref. [31], with (2) quantum metric contributing the interband term to the Kubo velocity-velocity correlator (in electronic case current-current correlator equivalent to the Kubo conductivity [J. Phys. Condens. Matter 30 414001 (2018)]), and (3) Kubo conductivity and diffusivity being in a direct correspondence [Phys. Rev. Lett. 85, 2422]. Similar relations concerning localisation and quantum metric were previously established in the electronic context [Phys. Rev. B 56, 12847], analogously to the here retrieved relation, which we originally showed for the excitonic states in the Supplementary Information (SI), following the previous result of Ref. [31]. To clarify the origin of the equation, we update the referencing of the mentioned underlying works underpinning this equation in the main text.

4. A related question: Despite the fact that Eq. 5 gives a lower bound of the diffusion constant for the topological state in the flatband limit, it's worthwhile to compare the energy dispersion contribution and the quantum geometric contribution to the diffusion constant. If the quantum geometric contribution is a higher-order effect compared to the energy dispersion part, the bound becomes less useful.

We thank the Referee for their comment. We agree that Eq. 5 indeed gives a lower bound. We show in Figure 2 the explicit resolution in band (dispersion) and quantum geometric contributions to the overall diffusion for a range of parameter values. This is also demonstrated pictorially in the Supplementary Information (SI), where we show the diffusion for $n = 7$ (polyheptacene), which is known to have relatively flat exciton bands. We thereby compare the case with geometry [Fig. S1(e)] and without geometry [Fig. S1(f)]. As we show, in this case, the diffusion is indeed driven by this lower bound on the quantum geometric contribution and is not a higher order effect.

5/12

We draw attention to these details with the following changes in the main text:

Additional material-specific data is shown in the Supplemental Material (SM) Fig. S1, which further elucidates the importance of the geometric contribution, particularly for polyheptacene, which has a significantly flatter band structure.

5. What numerical method is used? Across the manuscript, it seems only the broad term "numerical simulation" is mentioned. Without the specific numerical methods and some detailed parameter specification, the reliability of the numerical results is doubtful.

We thank the Referee for their comment and for highlighting this. We have indeed used the catch-all term "numerical simulation" (or related) to describe the fact that we numerically evaluate the equations throughout the manuscript with realistic parameters.

Our calculations of excitons follows the methodology of our previous work, as outlined in the text. We use the Wannier equation combined with screening models aimed at capturing the dielectric environment, parametrised via first principles density functional theory (DFT) calculations of electrons. We can then extract the excitonic properties including exciton wavefunctions and bandstructures. For example, where the precise term "numerical simulation" appears in the text, we are referring to numerically calculating the exciton group velocity.

To address this, we change the language throughout the text from generically referring to simulation (which is ambiguous) to more concrete, detailed descriptions, as highlighted in red. We also explicitly discuss the input parameters in the updated Supplementary Information (SI) [Sec. I in the updated SI], extending the first principles to include our numerical methods for calculating excitons.

6. The discussion on exciton-polarons—where excitons become "dressed" by lattice distortions—is somewhat contradictory. The text claims that topological excitons are more "robust to polaronic effects." However, the data shows that topological exciton-polarons exhibit a larger energy shift and a greater reduction in velocity compared to their trivial counterparts, precisely because of the stronger exciton-phonon coupling. The plot in Figure 5(d) also shows a more significant velocity reduction ($V_{Exc}/V_{Exc-Pol} > 1$) for the topological case at low momentum. The claim of "robustness" seems to refer to the fact that the system's topological invariants are not destroyed by polaron formation, not that the transport properties are less degraded. This phrasing is confusing and could be misinterpreted.

We thank the Referee for their comment and agree the previous statement could be somewhat misleading. The Referee is correct – we are claiming that the topology of the underlying excitons is unchanged by polariton formation. Indeed, we do clearly observe an enhancement in the polariton shift and larger band renormalisation, owing to the enhanced electron-phonon coupling. To address and clarify this point, we update the main text by writing:

As discussed earlier, the stronger exciton-phonon coupling experienced by topological excitons results in higher dephasing rates, but we find that these are not sufficient for the topological excitons to violate the original quantum geometric bounds of the free exciton propagation, as compared to the trivial excitons. Similarly, the metric-enhanced transport of topological excitons still persists in the polaronic regime.

6/12

7. How does disorder affect the transportation in such a system? Will it be so dominating that the enhancement of the topological excitation becomes negligible?

The Referee raises an interesting point. Disorder and defects are particularly interesting. Generally, disorder is just another scattering mechanism, which gives rise to, for example, increases to the exciton line-width and a slow-down of the exciton transport. Importantly in our systems, the interplay between the exciton velocity and the exciton-defect scattering is likely to still favour enhanced topological exciton transport, owing to the quantum geometry. In certain cases, excitons can also become localised at defect sites, interesting for many applications including single-photon emission. In the context of topology, as discussed in our reply to the Referee's first comment, exciton defect/interface interactions are likely to be highly interesting in terms of excitonic edge states, solitons and exciton localisation, all of which we predict could be impacted by topology, following lessons from electronic systems. This is beyond the scope of our current work.

Nevertheless, it should be noted that defects were recently found to induce enhancements in the electronic quantum geometry [Phys. Rev. Lett. 133, 026002 (2024)], which transferred to excitonic geometry under interactions, could result in similarly promising enhancements, which deserves a separate detailed study. We believe that our findings are robust and provide an accurate description of exciton transport in the bulk polymer chain in the absence of defects and the general findings will also hold where defect scattering is relatively weak, i.e. with no sharp chemical interfaces or hard edges. To address this point, and highlight interesting possibilities opening avenues for future promising results, which could follow our geometric exciton transport result, we expand the Discussion with the following statements:

Throughout this work, we do not consider the role of defects or interfaces, which would lead to additional scattering mechanisms [Physical Review Materials 3, 074004 (2019)] or, if pronounced enough, localised states [Nat. Commun. 7, 13986 (2016), Phys. Rev. Lett. 119, 046101 (2017), arXiv:2505.03343]. We assume a sufficiently large and clean system such that the optoelectronic behaviour is determined by the bulk excitons. Interestingly, some recent studies have shown defect-induced enhancements to electronic geometry [Phys. Rev. Lett. 133, 026002 (2024)], and how these could translate to the exciton picture will be the topic of future work.

7/12

Reviewer 2 (Remarks to the Author):

In this work, the authors explore a novel approach to enhancing exciton transport by leveraging topological excitons in organic materials. The study is clearly a continuation of their previous work published in the same journal [DOI: 10.1038/s41467-025-59257-5] (below I refer to it as paper 1). The analysis presented is both innovative and timely and is likely to be of considerable interest to the community.

We thank the Referee for their very positive feedback on the topic of our work and we are very pleased to learn about their constructive evaluation and recognition of its significance.

However, I believe the manuscript would benefit significantly from a stronger connection to application-relevant questions, experimentally accessible observables, and common organic systems. As it stands, the work feels too technical for the broad readership of Nat. Commun., and I would encourage the authors to provide more context and motivation to make the manuscript accessible to a wider audience.

We thank the Referee for their constructive criticism which we address point-by-point below.

1) What are the requirements for the formation of topological excitons? It would be helpful to include a brief explanation of the key structural and electronic characteristics needed to realise such excitonic states or systems. While part of this discussion is included in the authors' previous work (paper 1), revisiting these points here would make the present study more self-contained and accessible.

We agree with the Referee that to make the manuscript more self-contained, we should highlight the electronic characteristics needed to realise such systems. We believe the main recipe for realising topological excitons is quite simple. First, the system should possess topological electrons. Although not strictly necessary (as discussed in our first paper), we believe it is easier to engineer. Second, the system should be a bulk semiconductor – semimetallic materials for instance do not host excitons. Third, the system should have relatively weak screening such that excitons dominate, in bulk 3D inorganic materials the large screening leads to very small binding energies [AIP Advances 3, 11 (2013)] such that their behaviour cannot be well resolved. In principle, therefore, we believe that there are a wide range of systems which could host topological excitons. To address this, we add the following to the main text:

Following our previous work [Nat. Commun. 16, 4661 (2025)], we found that the excitons inherit their non-trivial topology from the underlying electronic topologies rather than through interaction effects [Phys. Rev. Lett. 133, 176601 (2024)]. The quasi-1D nature of these crystals and the weak dielectric screening of organic molecules lead to large excitonic binding energies, which, combined with the large band gaps, ensures that these topological excitons dominate the optical response.

2) Is it truly necessary to use quasi-metallic polymers (such as acenes connected by carbon-carbon bonds) to realise this topologically enhanced transport? In other words, does the system need to be very conductive in order to maintain topological features? I am asking this because in Fig. 4 the diffusion coefficient both in topological and trivial exciton are very large (at least one to two orders of magnitude larger than in common van der Waals organic crystals). In my view, this raises the question about whether the novel physics highlighted in this work is only applicable to quasi-metallic systems, rather than to a broader class of organic materials as the authors seem to suggest. Could the authors clarify this and add some discussion?

8/12

We thank the Referee for their questions. We agree that (1) the exciton diffusion is relatively large in these systems and that (2) this stems (in part) from the dispersive electronic band structures due to the carbon-carbon bonds. The main motivation for choosing the semiconductors, which can be turned into “quasi-metallic” system was due to the tunability. There is experimental evidence showing that the polyacene chains explored in our work can host topological electrons [Nat. Nanotechnol. 15, 437–443 (2020)], and that by changing the chemistry of the acene building block the topology can be controlled. However, unsurprisingly, since these structures are covalently bonded, the band structure is much more dispersive than in common van der Waals bonded organic semiconductors.

To explore this further, we show in both the Supplemental Material and the updated Figure 4, that in the case the polyheptacene, the topological diffusion is still enhanced. Polyheptacene has a much flatter electronic band structure (as shown in our previous work [Nat. Commun. 16, 4661 (2025)]) which is more similar to those mentioned by the Referee. Crucially, Figure 4 shows that in the presence of phonons, the exciton diffusion constant of polyheptacene is still topologically enhanced, however the absolute values are smaller, particularly for the trivial case (see also response to Referee 1 comment 2).

Quantitatively, for the real topological system, the raw diffusion values at room temperature (300 K) are 1.76 cm²/s and 0.44 cm²/s for polypentacene and polyheptacene, respectively. In contrast, the artificial trivial states (made by swapping the order of t1 and t2 with the same underlying electronic band structure) have diffusion constants of 0.61 cm²/s and 0.103 cm²/s for bond-reordered polypentacene and polyheptacene, respectively. This brings our results more in-line with vdW bonded systems, however we should make it clear that our exciton diffusion is bounded from below, so it is not surprising that our results here still significantly exceed those in “conventional” organic semiconductors.

To highlight the generality of our results to less conductive systems (polyheptacene), we draw attention to polyheptacene:

Additional material-specific data is shown in the Supplemental Material (SM) Fig. S1, which further elucidates the importance of the geometric contribution, particularly for polyheptacene, which has a significantly less-dispersive band structure.

We also update Figure 4, to show both polypentacene and polyheptacene. We further add the following text:

To demonstrate the generality of the topological enhancement to the phonon-mediated exciton transport, we also compare the trivial and topological regime for polyheptacene. Due to the flatter bands, the exciton transport is reduced in both regimes. For the real topological system, the raw diffusion values at room temperature (300 K) are 1.76 cm²/s and 0.44 cm²/s for polypentacene and polyheptacene, respectively. In contrast, the trivial states have diffusion constants of 0.61 cm²/s and 0.103 cm²/s for bond-reordered polypentacene and polyheptacene, respectively. Here, the topological diffusion constant at room temperature is around 3 times larger than the trivial case for polypentacene, compared to around 4.5 for polyheptacene. This difference stems from the larger percentage contribution of the geometric to the excitonic group velocity due to the flatness of the exciton band. This suggests that geometry of the excitons,

rather than the geometry of the electron-phonon coupling, is the dominant practical factor for the transport enhancements. Given the characteristic energy scales driving these phenomena, we expect analogous effect interplays in the other optoelectronic systems realising dynamics dominated by excitons.

9/12

3) Regarding the electronic structure, I believe the quality and accuracy of their tight-binding SSH Hamiltonian stems from its ability to accurately fit the valence and conduction bands of the systems considered. However, in extending this approach to more complex systems, such as 2D and 3D materials or systems with more intricate band structures, do the author think a TB Hamiltonian would be accurate enough? how do the authors envisage modifying the Hamiltonian? A more detailed discussion of these future directions would be valuable. At present, the manuscript only briefly touches on this with a general statement on page 2: "the key findings of this work are generally applicable to the transport of topological excitons in any material and dimension." A deeper and more concrete discussion of the broader applicability would strengthen the manuscript.

We thank the Referee for their comment and for considering how to strengthen our manuscript. We believe a tight-binding Hamiltonian will often be sufficient to capture the essential physics, for example one could consider, as a starting point Haldane-like models of Chern insulators in 2D, or 2D generalisations of the SSH model, which include higher-order topological insulators (HOTIs). Longer range couplings such as next-nearest neighbour and so on could be included in these models. For more complex systems a combination of first-principles Wannierisation will allow us to develop tight-binding like Hamiltonians and determine topological quantities while for certain systems like moiré, effective models such as the continuum model have proven extremely successful in capturing the underlying topology. To address this, we added the following to the main text:

When extending to higher dimensions and more complex structures, finding accurate tight-binding models will become more challenging. However, established first-principles computational tools for electronic Wannierisation such as those regularly employed to describe electronic topology [Rev. Mod. Phys. 84, 1419–1475 (2012)], will still be sufficient here. We also point out that recent advances in excitonic Wannierisation [Phys. Rev. B 108, 125118 (2023)] could assist in the calculations presented here, from fully first-principles methods.

4) The SSH Hamiltonian in Eq. 2 appears to include only nearest-neighbor interactions, am I correct? However, excitonic couplings are generally long-range, and it is well known that the sign and magnitude of non-nearest-neighbor interactions can significantly affect both the spectroscopic and transport properties of molecular aggregates (see, for example, the work of Spano et al.). Could the authors comment on the potential role of long-range interactions in their model? Specifically, would the topological features and associated transport behavior be affected by the inclusion of such terms?

We thank the Referee for their comment. Yes, we indeed only include nearest-neighbour interactions in our work. This is because we find good agreement between the DFT bands and our SSH tight-binding Hamiltonian [Fig. 6, Nature Communications 16, 4661 (2025)]. We agree with the Referee that next nearest-neighbour interactions should add an additional layer of depth increasing our parameter space from just t_1 and t_2 , to include also longer range interactions. Referring directly to the work of Frank Spano, who has explored the interplay between long and short range coupling in van der Waals bonded organic crystals, this is driven by dipole-dipole coupling between excitons localised on individual molecules rather than excitons spread over multiple molecules due to wavefunction overlap, as in our system. Such dipole-like interactions will not be significant here, since the electronic states (molecules) are covalently connected.

Crucially our findings show that even if the transport is primarily driven by long-range hopping, e.g.,

10/12

dipole-dipole interactions, which happens in systems with relatively flat electronic bands, our findings still set a lower bound on the exciton transport for topological excitons. As a result, the topological exciton transport in these materials will still be enhanced. We agree that including longer range couplings will add a richer level of physics to future studies on exciton topology and geometry, however this is beyond the scope of this work.

We make the following changes to the text to highlight this:

In some organic materials, exciton transport is driven by long-range dipole-mediated hopping [Chemical Reviews 118, 7069–7163 (2018)] due to the absence of band-driven transport and strong molecular dipoles. Crucially, our findings state that the lower bound sets a minimum diffusivity on the band-driven exciton transport in these materials, provided the excitons are topological, meaning that the exciton transport will also be enhanced even in systems where the hopping transport would otherwise dominate. In the polyacene chains, the Wannier-like nature of the excitons means these longer range exciton transfers are negligible.

where we added the reference: [Chemical Reviews 118, 7069–7163 (2018)].

5) From my perspective is not quite clear how the author introduced temperature in their calculations and related polaronic effects. What are they computing exactly with electronic structure?

We thank the Referee for their comment. Throughout the manuscript, the temperature is introduced via the thermal population of excitons (which we assume to be thermalised to a Boltzmann distribution, as a limit of Bose-Einstein, given excitonic energies vs. temperature) and via the population of the phonon bath which we model via a Bose-Einstein distribution. We make the following changes to the Methods section to ensure this is clear.

β

Here, n_q describes the population of phonons in mode β and momentum q , which we model using a Bose-Einstein distribution assuming a thermalised phonon bath [Nanoscale 16, 8996–9003 (2024), 2D Materials 9, 025008 (2022)].

6) The authors mentioned that “Experimentally, the dynamics of exciton transport can be visualised with time-resolved photoluminescence [43, 56] or with transient absorption [57].” Is there any experimental evidence (maybe hidden) that demonstrates topological enhancement in materials, which the authors can refer more specifically to?

We believe that there indeed might be experimental evidence (indeed, hidden, as the Referee suggests) that could demonstrate topological enhancement of exciton transport in materials. The main challenge experimentally is isolating the role of the topology/geometry. This is why the material platform studied here is particularly potent, as it allows chemical changes to the topology without profound changes to the underlying structure, environment, chemical composition. Currently, to our knowledge, no such measurements (photoluminescence or transient absorption) have been performed on the polyacene chains discussed in our work.

11/12

Reviewer 3 (Remarks to the Author):

We thank the Referee for taking their precious time to review our manuscript, especially given their Early Career Researcher stage. For the detailed responses to Referee’s questions, please see responses to the other reports.

12/12

Version 1:

Reviewer comments:

Reviewer #1

(Remarks to the Author)

The authors have addressed all the questions in the report in detail, and the manuscript can now be published.

Reviewer #2

(Remarks to the Author)

The authors have thoroughly addressed my comments and enhanced the strength and quality of the paper. Therefore, I am happy to recommend it for publication.

Reviewer #3

(Remarks to the Author)

I co-reviewed this manuscript with one of the reviewers who provided the listed reports. This is part of the Nature Communications initiative to facilitate training in peer review and to provide appropriate recognition for Early Career

Researchers who co-review manuscripts.
